# Significant spatial gradients in new particle formation frequency in Greece during summer

Andreas Aktypis[1,2], Christos Kaltsonoudis[2], David Patoulias[2], Panayiotis Kalkavouras[3], Angeliki Matrali[1,2], Christina N. Vasilakopoulou[1,2], Evangelia Kostenidou[8], Kalliopi Florou[2], Nikos Kalivitis[4], Aikaterini Bougiatioti[3], Konstantinos Eleftheriadis[5], Stergios Vratolis[5], Maria I. Gini[5], Athanasios Kouras[6], Constantini Samara[6], Mihalis Lazaridis[7], Sofia-Eirini Chatoutsidou[7], Nikolaos Mihalopoulos[3,5] and Spyros N. Pandis[1,2]

[1]Department of Chemical Engineering, University of Patras, Patras, Greece
[2]Institute of Chemical Engineering Sciences, ICE-HT/FORTH, Patras, Greece
[3]Institute of Environmental Research and Sustainable Development, National Observatory of Athens, Athens, Greece
[4]Department of Chemistry, University of Crete, Heraklion, Greece
[5]Environmental Radioactivity and Aerosol Technology for Atmospheric and Climate Impact Laboratory, NCSR Demokritos, Athens, Greece
[6]Department of Chemistry, Aristotle University of Thessaloniki, Thessaloniki, Greece
[7]School of Chemical and Environmental Engineering, Technical University of Crete, Chania, Greece
[8]Department of Environmental Engineering, Democritus University of Thrace, Xanthi, Greece

*Correspondence to*: Spyros N. Pandis (spyros@chemeng.upatras.gr)

**Abstract.** Extensive continuous particle number size distribution measurements took place during two summers (2020 and 2021) in 11 sites in Greece for the investigation of the frequency and the spatial extent of new particle formation (NPF). The study area is characterized by high solar intensity, fast photochemistry, and has moderate to low fine particulate matter levels during the summer. The average $PM_{2.5}$ levels were relatively uniform across the examined sites. The NPF frequency during summer varied from close to zero in the southwestern parts of Greece to more than 60% in the northern, central, and eastern regions. The mean particle growth rate for each station varied between 3.4 and 8 nm h$^{-1}$, with an average rate of 5.7 nm h$^{-1}$. In most of the sites there was no statistical difference in the condensation sink between NPF event and non-event days, while lower relative humidity was observed during the events. The high NPF frequency sites in the north and northeast were in close proximity to both coal-fired power plants (high emissions of $SO_2$) and to agricultural areas with some of the highest ammonia emissions in the country. The southern and western parts of Greece, where NPF was infrequent, were characterized by low ammonia emissions, while moderate levels of sulfuric acid were estimated ($10^7$ molecules cm$^{-3}$) in the west. Although the emissions of biogenic volatile organic compounds were higher in western and southern sectors, they did not appear to lead to enhanced frequency of NPF. The infrequent events in these sites occurred when the air masses had spent a few hours over areas with agricultural activities, and thus elevated ammonia emissions. Air masses arriving at the sites directly from the sea were not connected with atmospheric NPF. These results support the hypothesis that ammonia and/or amines are limiting new particle formation in the study area.

## 1 Introduction

One of the key challenges in climate change is to quantify the exact role of atmospheric particles in the system. Atmospheric particles influence climate both directly, by absorbing or reflecting solar radiation, and indirectly, by becoming cloud condensation nuclei (CCN) and affecting the formation and characteristics of clouds (Seinfeld and Pandis, 2016). The aerosol net impact on the energy balance of the planet remains uncertain (IPCC, 2021). Although combustion and other sources can be a significant source of particle number in the atmosphere, new particle formation (NPF) is the dominant source of new

particles in many environments worldwide (Kulmala et al., 2004; Kerminen et al., 2018). Atmospheric nucleation and subsequent growth of the formed fresh nuclei to larger sizes is an important, and sometimes the dominant, source of CCN on a global scale (Spracklen et al., 2006; Merikanto et al., 2009; Gordon et al., 2017). Also, since the formed particles belong in the ultrafine range (<100 nm) they can have adverse health effects (Oberdörster et al., 2004; Politis et al., 2008).

         The first step of NPF is the formation of a critical cluster through the synergistic activity of various gaseous precursors

(Kulmala et al., 2001; Curtius, 2006). Although several compounds have been shown to participate in atmospheric nucleation, large uncertainty and often conflicting results exist regarding the mechanisms that drive the phenomenon in different areas. Sulfuric acid is the key component of the new particles in most environments (Kulmala and Laaksonen, 1990; Erupe et al., 2010; Lee et al., 2019). The binary sulfuric acid – water nucleation was the one of the first mechanisms studied (Kulmala and Laaksonen, 1990) but it does not explain the high nucleation rates observed in the troposphere (Weber et al., 1999). Both

laboratory experiments and ambient measurements have provided evidence that ammonia, amines, highly oxidized molecules and other organic and inorganic compounds can be involved in nucleation (Riipinen et al., 2011; Kirkby et al., 2011; Almeida et al., 2013; Kirkby et al., 2016; Lehtipalo et al., 2018). Despite this progress, the dominating new particle formation mechanisms in most areas remain uncertain.

         Quantifying the spatial extent of NPF events is important for understanding the impact of the formed particles on

climate. However, the available direct evidence is limited because simultaneous measurements in multiple locations are required. Regional NPF events, taking place in spatial scales exceeding 100 km have been reported (Kerminen et al., 2018). Bousiotis et al. (2021) analyzed measurements from 13 sites in Europe and found that regional events (occurring at both urban and rural sites) account for 5% of the events in Spain and up to 60% of the events in Germany. Berland et al. (2017) characterized the NPF occurrence and characteristics in three Mediterranean islands (Mallorca, Corsica, Crete) and found that

in 8 out of the 41 days of the campaign, NPF occurred in at least 2 out of the 3 sites. It is not clear if the events also covered the rather large marine areas between these islands. Further to the east, Baalbaki et al. (2021) found significantly higher NPF frequencies in Cyprus compared to the closest station, Finokalia, Crete about 670 km away (Pikridas et al., 2012). In another study in the Eastern Mediterranean by Kalkavouras et al. (2021), 6 months of measurements in Athens, Finokalia and Amman (Jordan) were analyzed. Only 5 concurrent NPF episodes were found in all three sites, all connected with northern air masses.

Meanwhile, local NPF and growth (scales of 10s of km) can take place inside plumes of combustion sources, in coastal sites

or urban areas (Kerminen et al., 2018). Kammer et al. (2023) estimated that the NPF spatial scale in an agricultural site in France is 20 km or a little more.

The few previous relevant studies in Greece suggest significant spatial variability of the NPF frequency during summertime. Patoulias et al. (2018) reported that during June 2012 the measured NPF frequency at Eptapyrgio, Thessaloniki in northern Greece, was close to 80%. However, measurements between June and October 2009 at the same location showed an event frequency close to 30% (Siakavaras et al., 2016). Athens is characterized by an intermediate to low NPF frequency, varying from 10 to 30% during summertime months (Petäjä et al., 2007; Vratolis et al., 2019; Kalkavouras et al., 2020). In Patras, in northwest Peloponnese, although NPF is infrequent (about 10%) (Patoulias et al., 2018; Aktypis et al., 2023), during 30% of the days new particles appear at the site, which had been formed upwind 100-150 km northeast of the city (Aktypis et al., 2023). In NEO, in southwest Peloponnese, a low frequency of around 5% during the summer has been found by Hansson et al. (2021). Furthermore, in the island of Crete, Kopanakis et al. (2013) reported only 2 NPF events in 54 summer days (3%) in Chania, which is situated in the northwestern part of the island. Several studies in the Finokalia station, also in Crete but in the northeastern part of the island, suggest that the NPF frequency varies from 10 to 30% during summertime (Pikridas et al., 2012; Kalivitis et al., 2019; Kalkavouras et al., 2020). The spatially variable NPF in Greece during the warm period suggested by the above studies makes the country an excellent natural laboratory to study factors that may be limiting or enhancing the phenomenon in different areas. Pikridas et al. (2012) found that in Finokalia, NPF events occurred only when particles were neutral, while the non-events related to more acidic particles. A base, probably ammonia, appears to control NPF in that region (Pikridas et al., 2012). The role of ammonia in NPF was also investigated in Patras (where the events are infrequent) using a dual chamber system (Jorga et al., 2023). After filling both chambers with ambient air, ammonia was added in one of the two chambers of the system. The addition of ammonia resulted in NPF in the perturbed chamber in three quarters of the conducted experiments, indicating that indeed it could be the missing ingredient for atmospheric NPF in Patras during summertime (Jorga et al., 2023).

In this study we quantify the NPF frequency during two consecutive summers (2020 and 2021) in 11 different locations in Greece. This represents a high spatial density of measurements considering the small size of the country (scale around 800 km). We focus on the summer periods because conditions favorable for NPF (high solar radiation, lower condensation sinks, higher biogenic volatile organic compound emissions (Nieminen et al., 2014; Bousiotis et al., 2021) are present. This makes summertime an excellent period to investigate factors limiting new particle formation, because a high NPF frequency is expected everywhere. We take advantage of the significant spatial gradients in the frequency, behavior, and characteristics of the events in an effort to gain insights about the dominant new particle formation mechanisms in the Eastern Mediterranean and other similar regions.

## 2 Materials and methods

Continuous particle number size distribution measurements took place during two summers (2020 and 2021) as a part of the PANACEA project in 11 different locations in Greece. This coordinated effort included measurements in 6 of the largest Greek cities; Athens, Thessaloniki, Patras, Xanthi, Ioannina, Chania, a coastal site near a small village, Methoni (only in 2020), three remote areas; Finokalia, Lesvos (only in 2021), and Sifnos, and one elevated remote site (Mt. Helmos). The measurements took place between August 1 and September 30 during the 2020 summer campaign and between July 15 and September 15 during the 2021 summer campaign.

### 2.1 Sampling sites

The measurement sites were chosen to cover most of the country (Fig. 1) and to also represent different types of environments. The sites were classified as urban (URB), suburban (SUB), rural (RUR) and remote (REM) according to the criteria proposed by Larssen et al. (1999). In addition, they were also further characterized as coastal, continental, or high-elevation sites depending on their location. Detailed descriptions of the sampling sites are provided below.

#### 2.1.1 Athens (ATH[SUB])

Athens is the capital and largest city of Greece, with a population of approximately 4,000,000 inhabitants. The city is densely populated and is surrounded by mountains in the western, northern, and eastern sectors, facing the Saronic Gulf in the south. The measurements in Athens were performed at the National Centre for Scientific Research, Demokritos, (37° 59'42" N 23° 48'57" E, at 270 m a.s.l.) a suburban background site located about 8 km northeast of the city center. The station is located at the foothills of Mt. Hymettus in a pine forest area and is affected by transported pollution from the city, especially during daytime hours (Eleftheriadis et al., 2021). The station can also be influenced by traffic emissions from a highway 400 m to the southeast and a major road 1.5 km to the north.

#### 2.1.2 Thessaloniki (THES[URB])

Thessaloniki is the second largest city of Greece, with a population of approximately 1,100,000 inhabitants. The city faces the Thermaic Gulf in the southwest while it is surrounded by Mt. Hortiatis in the east and the north. The western part of Thessaloniki is flat and one of the largest agricultural areas of Greece begins 15 km from the city center. The measurements were conducted at the urban monitoring station of the Aristotle University of Thessaloniki (40° 38'15" N 22° 56'30" E, at 43 m a.s.l) located at the city center. Traffic emissions from a nearby major road (Venizelou street) influence the site (Siakavaras et al., 2016). Other notable anthropogenic activities include an industrial zone 15 km to the northwest and a cement plant 7 km north of the sampling site. More information about the site can be found on Siakavaras et al. (2016) and Samara et al. (2014).

### 2.1.3 Patras (PAT[SUB])

Patras is the third largest city of Greece, with a population close to 300,000 inhabitants and is located at the foothills of Mt. Panachaiko facing the Patraikos gulf. The measurements were conducted at the Institute of Chemical Engineering Sciences (ICE-HT). This is a suburban background site located approximately 8 km NE of the city center of Patras (38° 17'54" N 21° 48'34" E, at 100 m a.s.l.). The site is surrounded by low vegetation and olive trees, while notable nearby human activities include the Patras–Athens highway (1 km away), a cement plant (6 km away) and the port of Patras (11 km away).

### 2.1.4 Ioannina (IOA[SUB])

Ioannina is a city in northwestern Greece, with a population of 110,000 inhabitants. The city is surrounded by mountains, forming a plateau, and lies on the western shore of lake Pamvotis. The suburban sampling site of Ioannina was at the University of Ioannina (39° 36'55" N, 20° 50'12" E, at 500 m a.s.l.), located 6 km south of the city center. The area is characterized by low vegetation and the major road leading to the city center is 2 km east of the sampling site. Due to the combination of its

basin-like characteristics and the existence of a lake next to it, the city has a unique microclimate. It is characterized by calm winds and stable conditions and high relative humidity resulting often in fog events (Pilidis et al., 2005; Houssos et al., 2012).

### 2.1.5 Thrace (THR[RUR])

Xanthi is a city in northeastern Greece with a population of around 70,000. The city lies on the foothills of the Rodopi mountains (on the north and northwest) while the land is mostly flat in the other directions. The Aegean Sea is 20 km to the

south of the city. Measurements were conducted in Xanthi only during the summer of 2020 in the rural-periurban site in the Democritus University of Thrace (41° 09'00" N, 24° 55'12" E, at 75 m a.s.l.), located 4 km east of the city center. The nearby area is characterized by low vegetation while a major road is located 500 m to the south. The measurements in Thrace during the summer of 2021 were conducted in Xylagani, a village with a population of 1,200 inhabitants located about 45 km to the southeast of Xanthi. For this study we implicitly assume that the NPF events observed in both locations were representative of

the broader Thrace region and do not differ significantly from each other. Although the two sites can have significant differences regarding local ultrafine particle emissions, the present study focuses on NPF events (lasting for >2 h) that take place on a spatial scale larger than the distance (45 km) between them. In addition, the fine PM concentration levels of the two sites were compared and were found to be similar. The HYSPLIT analysis revealed that both areas are generally affected by the same air masses (mostly north and northeast) which is important when studying NPF.

### 2.1.6 Lesvos (LES[REM])

Lesvos is an island in the northeastern Aegean Sea, located 10 km from the western coast of Turkey. It is the third largest Greek island with a population of about 110,000 inhabitants. The measurements during the summer 2021 campaign were performed in a remote site 7 km from the village Sigri (39° 13'52" N, 25° 56'08" E, at 636 a.s.l.), in the western part of the

island. The village has a population of 300 inhabitants. The nearby area is characterized by low vegetation and a small road 200 m to the north. Lesvos is one of the sites that is largely affected by the Etesians which dominate Greece, and especially the Aegean Sea during the summer (Tyrlis and Lelieveld, 2013; Dafka et al., 2016).

### 2.1.7 Sifnos (SIF[REM])

Sifnos is a small island in the southwestern Aegean, with a population of approximately 3,000 inhabitants. The measurement site was located on the eastern outskirts of the village of Exampela (36° 58'03" N 24° 43'48" E, at 220 m a.s.l.) where no cars are allowed. The nearest local traffic road is 250 m to the west. The site is characterized as remote and is directly exposed to the northern flow of the Etesians which dominate during the summer.

### 2.1.8 Finokalia (FIN[REM])

Finokalia is a small rural settlement (<10 inhabitants) located in the northeastern coast of the island of Crete. The atmospheric observation station of the University of Crete is located 3 km north of the village (35° 20'15" N 25° 40'11" E, at 250 m a.s.l.), facing the Aegean Sea. It is characterized as a regional background station and is representative of the Eastern Mediterranean. There are no notable anthropogenic activities (Mihalopoulos et al., 1997) within 15 km. The city of Heraklion (200,000 inhabitants) is located 50 km to the west.

### 2.1.9 Chania (CHA[URB])

Chania is a coastal city in the northwestern part of Crete, with a population of 60,000. The measurements in Chania were conducted at the urban background station of the Technical University of Crete in Akrotiri (35° 31′48″ N, 24° 03′36″ E, at 137 m a.s.l.). The station is located 5 km northeast of the city center and 2 km downwind of the sea. The nearby area is characterized by low vegetation, residential areas, and farms. A more detailed description of the Akrotiri station can be found in Lazaridis et al. (2008).

### 2.1.10 Methoni (NEO[RUR])

The measurements (only for the 2020 summer campaign) were conducted at the Navarino Environmental Observatory (NEO), a rural site located in the southwestern part of the Peloponnese (36° 49'20"N, 21° 42'14"E, at 30 a.s.l.). The site is located at Methoni, a village with 2,600 inhabitants. Since the site is situated 400 m from the sea, it is also considered a coastal site. Low vegetation and olive trees characterize the nearby area.

### 2.1.11 Mt. Helmos (HAC[REM])

The Helmos mountain Hellenic Atmospheric Aerosol and Climate Change station (HAC)[2] is a high-altitude station in Greece, operated by the Environmental Radioactivity Laboratory of National Centre for Scientific Research (NCSR) 'Demokritos'.

The station is located at the peak of the Helmos mountain in Northern Peloponnese, (37° 59'02" N, 22° 11'45" E, at 2314 m a.s.l.). It is the only Greek station characterized as a free tropospheric background site (Collaud Coen et al., 2018).

## 2.2 Instrumentation

Scanning or differential mobility particle sizers (SMPS or DMPS) were used for the measurements in the various sites. At least one SMPS or DMPS system was deployed in each of the stations for the measurement of the particle number size distributions. Different models of classifiers, DMAs and CPCs were used in each site (Table 1) covering at least the diameter range from 14 to 430 nm.  During the summer 2021 campaign in Patras, the use of a second SMPS (classifier model 3080, CPC model 3775, DMA model 3085, TSI) together with a particle size magnifier (PSM), allowed the measurement of the aerosol concentrations

and size distribution down to 1.3 nm.

Supporting measurements of black carbon (aethalometer model AE33 and/or MAAP), $NO_x$ and/or $SO_2$ were also performed in several stations (Table 1) and were used for the interpretation of the origin of the nucleation mode particles especially in urban and suburban stations like Athens and Patras. Measurements of several meteorological parameters were also performed during the two campaigns and included temperature, relative humidity, wind direction and speed, solar

radiation, and precipitation. The meteorological stations were located either on-site or a few kilometers away from the corresponding measurement site. For Thessaloniki and Thrace, the meteorological data were obtained from the National Observatory of Athens network that is described by Lagouvardos et al. (2017).

## 3. Data analysis

### 3.1 Event classification

The identification and classification of NPF events was performed according to the method proposed by Dal Maso et al. (2005). The whole available size range at each site was utilized for the classification of the days. Excluding Patras, the lowest detectable size in all stations varied between 9 and 14 nm, which (although it introduces some uncertainty) makes the size distributions comparable for the classification step. In Patras, the classification was performed twice: a) by accounting both the full-size distributions and b) accounting only the ones for the size range 14 – 700 nm. The number of the NPF events in Patras was the

same with both methods. Information about the sub-10 nm particles in Patras was valuable for the identification and interpretation of the undefined events, observed frequently in that area (Aktypis et al., 2023). The main criteria for an episode to be classified as NPF include: a) the appearance of a distinct particle mode with diameter below 25 nm (nucleation mode), b) its persistence for at least 1 h and c) its subsequent gradual growth towards larger sizes. NPF events were further classified into: a) Class I, if during the event the evolution of particle concentration and size could be observed clearly and the dynamic

properties (i.e. particle formation and growth rates) could be estimated with high confidence level, and b) Class II if fluctuations in nucleation mode particle concentration and/or diameter were observed. Weak (events when new particles grew in size, but

either their number concentration was quite low or their growth was limited) or short-lived events (but still lasting for > 1 h) were also classified as Class II in this study. Days that did not meet the above criteria, but during which the appearance of nucleation mode particles was evident, were classified as undefined. These cases included days where either the growth of the observed particles was insufficient, or their initial formation was not observed, even if they exhibited some growth.

Supporting measurements of BC, $NO_x$, and/or $SO_2$ were inspected (when available) to help identify the origin of the nucleation mode particles observed during a potential NPF event because primary particulate matter (PM) emissions can often cause biases in the classification process, especially in urban and suburban areas. Days when nucleation mode particles were observed at the same time with significant increases in concentrations of black carbon or $NO_x$ or $SO_2$, were classified as non-events (although a few of them could be events, affected by local anthropogenic sources and photochemistry (Dai et al., 2017)). However, if during an event, several concentration peaks from nearby sources obscured the NPF event but still the formation and growth of new particles was evident (Fig. S1), the day was classified as a NPF event (Kulmala et al., 2012). Although great effort was made to accurately classify all days, due to the subjectivity involved in the method, small uncertainties in the final results are expected.

## 3.2 Air mass origin

The origin of the air masses was investigated calculating the 24-h backward air trajectories that arrived at each measurement station during the two summer campaigns. One trajectory was estimated for each hour (24 trajectories per day). The calculation was performed using the Hybrid Single-Particle Lagrangian Integrated Trajectory (HYSPLIT-4) model developed by the NOAA Air Resource Laboratory (Draxler and Hess, 1998) in the backward trajectory mode (Stein et al., 2015). A default arrival height equal to 50 m above ground level was chosen. For both summers, the NOAA Global Forecast System (GFS 0.25 degrees, NOAA) meteorological files were used. The Weather Research and Forecasting (WRF) model (Skamarock et al., 2008) was also used for the calculation of the meteorological fields for the summer of 2021, providing a more detailed perspective of the air mass circulation over Greece. WRF was used with a 4 km grid resolution for Greece that was nested into a 12 km grid for the rest of Europe. The calculated trajectory endpoints were further analyzed using the Openair package (Carslaw and Ropkins, 2012).

## 3.3 Calculation of the condensation and coagulation sinks and the growth rate.

The condensation sink (CS) is an important parameter, affecting the rate with which vapors can condense on pre-existing atmospheric aerosol (Kulmala et al., 2001). It can be calculated according to:

$$CS = 2\pi D \sum_i \beta_i d_{pi} N_i ,$$ (1)

where $D$ is the diffusion coefficient of the condensable vapors, $\beta_i$ the transition regime correction factor (Sutugin and Fuchs, 1970), $d_{pi}$ the diameter of the SMPS size-channel $i$ and $N_i$ the number concentration of that size-channel. The diffusion

coefficient of $H_2SO_4$ and a mean free path equal to 123 nm were chosen for the calculations in this work (Erupe et al., 2010; Dinoi et al., 2021).

The growth rate (GR) of the newly formed particles during an NPF event was calculated following the method described by Kulmala et al. (2012). Initially, the log-normal particle number size distributions were fitted using a least-squares fitting algorithm, and the geometric mean diameters of the nucleation, Aitken, and accumulation mode particles were determined. Next, the growth rate was determined as the slope between the starting time and diameter ($D_{p1}$) of an event and the time when the particles reached the 25 nm mobility diameter limit ($D_{p2}$), according to:

$$GR = \frac{\Delta(25\,nm - D_{p_i})}{\Delta t}, \tag{2}$$

where $\Delta t$ is the time needed for particles with diameter $D_{p_i}$ to reach 25 nm. Because the smallest detectable particle diameters vary from site to site (Table 1), uncertainties are introduced in the direct comparison of the calculated growth rates. For the calculation of the growth rates, only the class I events were considered, because the evolution of the geometric mean diameter could be followed with a good confidence level. The starting time of each event was defined as the time when the observed nucleation mode geometric mean diameter started increasing.

## 4. Results and discussion

### 4.1 Ambient conditions during the measurements

Summer in Greece is characterized by relatively high temperature and sunlight intensity and relatively dry conditions. The temperature across all stations varied between 11 and 39 °C during the summer 2020 campaign, having an average of 25.8 ± 3.3 °C and between 16 and 41 °C during the summer 2021 campaign with an average of 26.8 ± 4 °C. The average temperature during the expected NPF occurrence periods (9:00-15:00 LT) was 28.5 °C. The summer 2021 campaign was characterized by some periods with extremely high temperatures due to a heatwave that peaked at the beginning of August (Founda et al., 2022). The average relative humidity of all stations was 60.1 ± 14.4% and 57.7 ± 14.8% for the two campaigns, respectively. The peak of the solar radiation exceeded 800 W m$^{-2}$ around noon during most days of the campaigns, while rain events were overall rare (less than 5, mostly occurring at the end of September). Average meteorological parameters for the two measurement periods and each site can be found in Table S1 for 2020 and Table S2 for 2021.

The atmosphere was relatively clean in Greece during the measurement periods. Dimitriou et al. (2023) analyzed measurements of $PM_{2.5}$ from 14 Purple Air sensors in Athens, Thessaloniki, Patras, Xanthi and Ioannina between July 2019 and June 2021 and found that the average $PM_{2.5}$ levels were low (with a small variation from 8 to 10.8 μg m$^{-3}$) during the summer. Even more interesting was the relatively homogeneous distribution of $PM_{2.5}$ across the country. This suggests that, with the exception of the free tropospheric station of Helmos, the other 10 had similar $PM_{2.5}$ and sunlight levels.

The average $N_{14}$ (mean number of particles with diameters larger than 14 nm calculated for both campaigns) concentrations varied significantly (Fig. 2). Helmos[REM], Chania[URB], Sifnos[REM], Finokalia[REM] and NEO[RUR] had the lowest average $N_{14}$ ranging from 2,000 to 3,000 cm$^{-3}$. Somewhat higher number concentrations were observed in Patras[SUB], Lesvos[REM], and Athens[SUB], where the average $N_{14}$ varied between 3,000 and 4,000 cm$^{-3}$. Athens had higher than average nucleation mode

particles mainly due to primary emissions (Eleftheriadis et al., 2021). Thrace[RUR] had an average $N_{14}$ of about 4,170 ± 3,200 cm$^{-3}$, while in Ioannina[SUB] the $N_{14}$ was 6,910 ± 3,200 cm$^{-3}$. Finally, Thessaloniki[URB] had by far the highest average $N_{14}$ compared to the other sites, equal to 10,380 ± 5,900 cm$^{-3}$. The high particle concentration observed in Thessaloniki can be explained by the urban characteristics of the measuring station and to the frequent and intense NPF events identified in that site. Except Ioannina, the average $N_{14}$ of the two campaigns in each station were quite similar.

The nucleation mode ($N_{14-25}$) particles accounted on average for 12 ± 9% (the latter number represents one standard deviation from the mean) of the total particle number concentration ($N_{14}$), with the corresponding fraction varying from 3% in Chania to 35% in Thessaloniki. The Aitken mode ($N_{25-100}$) particles accounted on average for 54 ± 5% of $N_{14}$ varying from 47% in Thessaloniki to 62% in Lesvos. Finally, the accumulation mode ($N_{100}$) particles corresponded to 34 ± 8% of $N_{14}$ varying from 18% in Thessaloniki to 45% in Chania. The $N_{14}$ and $N_{100}$ were calculated accounting for the upper size limit available in

each site. This ranged between 430 and 800 nm, but the number contribution of these particles in the total ($N_{430-800}/N_{14}$) was negligible (aways below 2%) compared to that of the smaller particles.

## 4.2 NPF frequency

A total of 941 daily particle number size distribution plots (average data coverage was 76%) were visually inspected together with the corresponding evolution of the nucleation, Aitken, and accumulation mode number concentrations in order to be

classified as Class I, Class II NPF events, undefined or non-events. Typical examples of each different category together with their corresponding $N_{14}$ number concentration are depicted in Fig. 3. The formation and growth of new particles is clear in the Class I NPF event in Sifnos on 14/8/2020 (Fig. 3a). Newly formed particles appeared at around 10:00 LT and grew for 6 hours reaching sizes of 100 nm. During the Class II event (Fig. 3b) in Finokalia on 23/7/2021, new particles were observed at 8:00 LT but the evolution of their diameter and number concentration were not continuous. In the undefined event shown in Fig.

3c in Patras on 27/7/2021, new particles with a mean diameter of 25 nm appeared at around 11:00 LT, but their formation and their growth were absent. This day was classified as a transported event by Aktypis et al. (2023). Finally, Fig. 3d shows a day where no NPF was observed in Chania.

A significant spatial variability of the NPF frequency (calculated as the total number of Class I and II events divided by the total number of days with available data) was observed across the 11 different areas during both summer campaigns

(Fig. 4). The average NPF frequency varied from close to zero in NEO and Chania to 64% in Thessaloniki. Although a substantial variation of the frequency in a given station from year to year is possible (Dal Maso et al., 2005; Manninen et al.,

2010, Kalivitis et al., 2019), no significant differences (less than 10%) were observed in the Greek stations between the two campaigns, except Finokalia.

The three stations (THES[URB], LES[REM], THR[RUR]) in the north and the northeast had all intense events and high NPF frequencies (defined here as more than 40%). Thessaloniki had the highest NPF frequency (64%) compared to the other sites. The frequency was higher during the first summer (68% vs. 61%), due to more class I events (40% vs. 27%). Lesvos had a NPF frequency of 60% during the summer 2021 campaign, with 40% being Class I events. Thrace had a frequency of 45% during 2020 and 49% during 2021, while the Class I events were 24% and 20%, respectively.

Sites with intermediate NPF frequency (between 20% and 40%) included Sifnos[REM], Finokalia[REM] and Helmos[REM]. The frequency of NPF events at Sifnos was 33% during 2020 and 36% during 2021, with the class I events accounting for 14% and 16%, respectively. A considerable difference in the event frequency between the two campaigns was observed in Finokalia, where NPF was 18% (class I, 7%) and 35% (class I, 17%), respectively. During 2021 the air in Finokalia was dryer (Tables S1 and S2) and the wind speed was higher than 2020. Those significant meteorological differences could explain the increased NPF frequency in Finokalia during 2021, compared to 2020. NPF events in Mt. Helmos occurred during 30% of the summer days in 2020 and 27% in 2021, with the class I events only occurring at 12% of all days in both campaigns.

Sites located mainly in the western and southern parts of Greece (Patras[SUB], Ioannina[SUB], NEO[RUR], Athens[SUB], and Chania[URB]) were characterized by low NPF frequency (<20%). In Patras the event frequency was 15% during 2020 and 10% during 2021, while the Class I events were only 3 in total (Aktypis et al., 2023). The corresponding values for Ioannina were 17% and 11%, while the Class I events were also rare (only 2 in total). The NPF frequency in Athens was 14% during 2020 and 9% during 2021, while only 3 Class I events were observed exclusively during the first campaign. In Chania the frequency was 9% and 3% during the two campaigns and all events were Class II with relatively low concentrations of formed particles. NEO (during summer 2020) had the lowest NPF frequency, with only two (4.5%) Class II events occurring during the 45 days of the 2020 summer campaign. Nighttime bursts of nucleation mode particles that grew to larger sizes were occasionally observed in NEO and will be analyzed in the next section. An example is depicted in Fig. S2, and these bursts were not counted as NPF events.

## 4.3 NPF event characteristics

In this section, various characteristics of the events like their possible relation with meteorological parameters, the condensation sink as well as the growth rates are analyzed for each station. Meteorological data were compared, and the CS calculations were performed during daytime, and more specifically in the time window between 8:00-18:00 (LT), when NPF generally occurs. A t-test was performed to compare the CS, RH and wind speed during the events and the non-events. The 99% confidence interval was chosen to reduce uncertainty in the comparisons. To evaluate the assumption that the data follow a normal distribution, the corresponding frequency and QQ plots were analyzed. Excluding outliers all distributions were close to normal.

In most sites, NPF took place during days with lower relative humidity. The average RH was significantly (a=0.01)
lower during NPF days in 6 sites (CHA[URB], FIN[REM], LES[REM], THES[URB], THR[RUR], NEO[RUR]) (Fig. S3). Lower RH during NPF
has been observed worldwide (Kerminen et al., 2018), as well as in the Eastern Mediterranean (Kopanakis et al., 2013;
Kalkavouras et al., 2020). There was not a clear relation between wind speed and NPF events (Fig. S4). In 5 sites (PAT[SUB],
FIN[REM], SIF[REM], THES[URB], THR[RUR]) the wind speed was significantly higher (a=0.01) during NPF days, while the opposite
was observed in 2 sites (ATH[SUB], CHA[URB]). In the rest, there was not a statistically significant difference. The corresponding
ambient temperatures during events and non-events were also compared (Fig. S5). For most sites (THR[RUR], SIF[REM], FIN[REM],
NOA[RUR], PAT[SUB], IOA[SUB]) the temperature during event days was statistically the same compared to the non-event days. In
THES[URB], LES[REM] and CHA[URB] the temperature was on average a little higher (less than one degree) during NPF events, while
the opposite was observed for HAC[REM] and ATH[SUB]. This analysis suggests that temperature did not play a dominant role in
determining the occurrence of NPF.

NPF events typically are associated with lower values of the condensation sink (Kerminen et al., 2018), because pre-
existing aerosol absorbs quickly the available condensable vapors, preventing this way the formation and growth of small
clusters. The calculated condensation sinks were compared between the event, non-event, and undefined days for each location
(Fig. 5). In most of the sites there was not a considerable difference of the CS between events and non-events. This behavior
has also been observed in Cyprus in the Eastern Mediterranean by Baalbaki et al. (2021). Only in 4 out of 11 stations there
was a significantly (a = 0.01) lower CS during NPF events compared to the non-events. Three of these sites were in the western
part of the country (Patras, NEO, and Ioannina) and the last one was Lesvos. Interestingly, in Thessaloniki, the numerous
events were associated with a slightly higher CS than the non-events. Siakavaras et al. (2016) found that NPF in Thessaloniki
occurs frequently during days with high levels of preexisting aerosol. This may have to do with the origin of the air masses
(analyzed in the next section) that arrive in Thessaloniki and can affect both NPF and PM concentrations. Overall, the
intercomparison between the sites suggests that the average CS did not vary a lot in most locations and was relatively low
during the measurement periods (around $6 \times 10^{-3}$ s$^{-1}$). This uniformity of the CS is consistent with the PM$_{2.5}$ observations of
Dimitriou et al. (2023) during the same periods. Thessaloniki and Ioannina were two exceptions with an average CS of 14 ×
$10^{-3}$ s$^{-1}$ and 19 × $10^{-3}$ s$^{-1}$, respectively. In Thessaloniki, the higher average CS can be explained by the central urban location
of the site. The city of Ioannina was characterized by stagnation of the air (average wind speed was 1 m s$^{-1}$) during the study,
but this alone cannot explain the high CS observed there. It is probably due to the combination of the low wind speed with the
significant local sources that leads to the high average CS. This is not the case in other sites with comparable wind speeds like
Thrace.

The average growth rate of all the sites was 5.7±2.9 nm h$^{-1}$ and varied from 3.4±1.8 nm h$^{-1}$ in the high elevation site,
Helmos, to 8.0±1.6 nm h$^{-1}$ in Finokalia. Although there was a significant difference between these two locations, overall, the
average growth rate in the rest of the sites (Fig. 6) did not have a significant variability. The average growth rate observed is
consistent with previous observations in the eastern Mediterranean. In Helmos, the relatively low growth rates could be

explained by the limited availability of condensable vapors in this high elevation area (Sellegri et al., 2019). On the other hand, the high growth rates observed in Finokalia imply high levels of vapors that condense rapidly. Relatively high growth rates in Finokalia during the warm period have also been reported in previous studies. Kalivitis et al. (2019) found that the 10-year average of the growth rate during the summer in Finokalia was equal to $7.3 \pm 3.9$ nm h$^{-1}$. Even higher growth rates during summer have been reported by other relevant studies in this location (Pikridas et al., 2012; Berland et al., 2017; Kalkavouras et al., 2020). The potential linkages between the growth rate and meteorological conditions (T, RH, wind speed) for the sites that had enough growth rate data points (THES, THR, FIN, SIF, HAC, LES) were investigated. We did not observe any systematic dependency of the GR on any of these parameters. This suggests that other non-meteorological factors control the growth rate in these regions. The observed NPF frequencies, and the growth rates are comparable with the ones reported for other areas in Europe during the summer by Nieminen et al. (2018). This also includes the small growth rates observed in the high-altitude sites which were also observed at Helmos.

## 4.4 Emissions of SO$_2$, NH$_3$ and biogenic volatile organic compounds in Greece during summer

Since the NPF frequency and intensity are largely dependent on the availability of the various gaseous precursors involved in nucleation (H$_2$SO$_4$, NH$_3$, volatile organic compounds etc. (Kerminen et al., 2018; Lee et al., 2019), it is important to investigate their emission levels and spatial distribution across the country. Sulfur dioxide and ammonia emissions for the summer of 2020 were based on the CAMS-REG emission inventory (Kuenen et al., 2022) for the summer of 2018. We assume here that the emissions and especially their spatial distribution did not change drastically in this short period. these years. Biogenic emissions were estimated by the Model of Emissions of Gases and Aerosols from Nature Version 3 (MEGAN3) (Guenther et al., 2012, 2020).

Sulfur dioxide (SO$_2$) levels can be a good indicator of sulfuric acid during summer in Greece. Significant amounts of SO$_2$ (Fig. 7a) are emitted in the Balkans (European Environment Agency, 2022) by coal or lignite-fired power plants. Although a significant reduction (almost 80%) of SO$_2$ emissions has taken place in Europe between 2005 and 2020 (European Environment Agency, 2022), relatively high emissions are still present in the Balkans. The Maritsa Iztok complex in eastern Bulgaria for example is one of the largest SO$_2$ emitters in Eastern Europe (Krotkov et al., 2016). Western Turkey also has elevated SO$_2$ emissions (Fig. 7a), due to the numerous power plants located there. Compared to these areas, Greece has lower SO$_2$ emissions. Elevated emissions in the Greek region are observed in the north, where the largest lignite-fired power plants of the country are located. Thus, higher SO$_2$ and therefore H$_2$SO$_4$ concentrations in the north are strongly associated with the highest NPF frequencies in the northern part of the country (i.e., THES$^{URB}$, THR$^{RUR}$). Although there are low SO$_2$ emissions in the southern and western parts of the country, these regions during summer have adequate sulfur levels, due to both long-range transport from the north and also ship emissions (Dayan et al., 2017). The available SO$_2$ measurements in Patras, revealed an average concentration of $0.42\pm0.27$ ppb, that results in adequate levels of H$_2$SO$_4$ (in the order of $10^7$ molecules cm$^{-3}$) for nucleation, at least in the western part of Greece. Sulfuric acid is assumed to be in pseudo-steady state in the gas phase with

its production rate being proportional to the sulfur dioxide concentration, the OH radical concentration and inversely proportional to the condensational sink (Bardouki et al., 2003; Pikridas et al., 2012). We implicitly assume here that the reaction of sulfur dioxide with OH is the dominant source of sulfuric acid in the study area. The levels of OH were estimated using the predictions of the chemical transport model PMCAMx (Fountoukis et al., 2011) for the summer, while the sulfur dioxide and condensational sink were measured.

Significant spatial gradients in ammonia emissions are observed in Greece (Fig. 7b), dividing the country into two sub regions. The central and northeastern parts are characterized by elevated emissions, exceeding 2 kg km$^{-2}$ d$^{-1}$ in the agricultural areas of Thessalia, Thessaloniki and Thrace. These are the largest agricultural areas of Greece, while livestock farming is also important there (Gemitzi et al., 2019; Hellenic Statistical Authority, 2021). Emissions of amines in these areas may also be elevated, since animal husbandry and agricultural activities are important amine and ammonia sources (Ge et al., 2011). The emissions in western and southern Greece are limited over smaller (mainly agricultural) areas such as Etoloakarnania, Ilia (northwest and southwest of Patras, respectively) and central Crete. An exception is Ioannina, where the ammonia emissions are elevated, but the NPF frequency was low. This could be partially explained by the increased mass concentration of preexisting particles in that area, that results in higher condensation sink (Fig. 5). The ambient CS can be further enhanced by the relatively high RH in the area due to the lake next to the city.

Contrary to ammonia, higher emissions of biogenic volatile organic compounds (Fig. 7c and 7d) are observed in the western and southern parts of the country. These regions are characterized by various mountain complexes (Fig. 1) and densely forested areas (Gemitzi et al., 2019). Kaltsonoudis et al. (2016) reported ambient concentrations of monoterpenes during the summer equal to 0.3 ppb and 0.6 ppb in the cities of Patras and Athens, while the average isoprene concentrations were equal to 1.0 and 0.7 ppb respectively. Monoterpene concentrations as high as 2.9 ppb and isoprene as high as 5 ppb have been reported in a forest in the Agrafa mountains, Northern Greece during summer (Harrison et al., 2001). Despite these relatively high emissions and concentrations biogenic organic vapors do not appear to enhance NPF frequencies in the western and southern parts of the country. Temperature has been directly connected with the ability of the highly oxygenated organic molecules to participate in NPF (Simon et al., 2020). The relatively high temperatures during the measurement periods may also be part of the explanation of why the organics do not appear to enhance the NPF frequency in their corresponding high emission areas. This can lead not only to more volatile oxidized organics but may also reduce the stability of small clusters in the atmosphere (Bousiotis et al., 2021). High isoprene to monoterpene ratios could also be part of the explanation, since they have been connected with suppression of NPF (Kiendler-Scharr et al., 2009; Kanawade et al., 2011; Lee et al., 2016).

## 4.5 Air mass origin during the NPF events

To gain insights about the potential nucleation mechanism(s) that dominate Eastern Mediterranean it is useful to investigate the air mass origin during the NPF events and compare it with the entire period and the non-event days (Wonaschütz et al.,

2015). This analysis is restricted to the time window between sunrise and sunset (7:00 - 19:00 LT), when all the NPF events occurred.

### 4.5.1 Northeastern Greece

All three sites in northeastern Greece (THES[URB], THR[RUR], LES[REM]) were dominated by northeastern air masses passing over the Eastern Balkans and Northwestern Turkey. Cluster analysis (Carslaw and Ropkins, 2012) was performed for the entire

measurement period (Fig. 8a) to group all trajectories depending on their origin. An angle distance matrix was used, which compared to the Euclidean, takes more into account the movement patterns and shape of the trajectories (Carslaw and Ropkins, 2012). The analysis revealed that during 44% of all days, northeastern air masses were prevailing in Thessaloniki. Air masses from the northeast also prevailed in Thrace (52%) and Lesvos (60%) during the whole period. Other important sectors included the northern and western Balkans and the south, where the corresponding trajectories passed at least 6 hours over the Aegean

Sea before arriving at each site. Air masses from the east were common mainly in Thrace.

Most of the NPF events (43%) in Thessaloniki were associated with air masses from the northern sector, while the corresponding fraction for the non-events was only 19% (and the whole period 27%). These trajectories (Fig. S6) passed over N. Macedonia, the Ptolemaida region (where the largest lignite-fired power plants of the country are located) and the agricultural area of Thessaloniki a couple of hours prior to arriving at site. The increased potential of these air masses to form

new particles can be explained by the fact that they are expected to be rich in both $SO_2$ and $NH_3$ due to the particular route they follow (Fig. 7a and 7b). A large fraction (38%) of the events was associated with trajectories from the northeast, usually passing over eastern Bulgaria or northwestern Turkey 10-20 h before arrival. These air masses also spent some time over the agricultural region of Thrace, located about 160 km to the east of the site. Air masses originating from the southwest were moderately associated with NPF in Thessaloniki.

Northeastern winds (Fig. S7) passing over Eastern Bulgaria were prevailing in Thrace during 57% of the NPF events, while the corresponding fraction for non-events was 33%. NPF under eastern air masses was also frequent in Thrace (34%). There was not a single event associated with southern wind directions, a condition present during 15% of the days.

In Lesvos almost all NPF events took place when the air masses were northeastern (Fig. S8), passing over western Turkey or eastern Bulgaria. Only two events occurred when air masses came from the south. Being downwind of the

northwestern Turkey region, the corresponding air masses are expected to be rich in $SO_2$ since several lignite-fired power plants are located there (Say, 2006; Vardar and Yumurtaci, 2010). During all five NPF events in Lesvos under northern air masses, NPF was also observed in Thrace on the same day. This suggests that these occurred on a spatial scale of more than 200 km.

### 4.5.2 South Aegean

The sites located in the south Aegean (SIF[REM], CHA[URB] and FIN[REM]) were exposed to the northeastern flow of the Etesians. During 44% of the measurement days, northeastern air masses were reaching Sifnos, while the corresponding fractions were 48% for Finokalia and 69% for Chania (Fig. 8b). Northern flows from the Balkans and western air masses mostly of marine origin were also observed frequently (25-50%).

Most of the NPF events in Sifnos (73%) occurred in fast moving air masses from the northeast (Fig. S9). Only three
events were associated with trajectories from the west, that in all cases had spent several hours over the Peloponnese. Not a single event occurred in southwestern air masses, which were common during non-events. During the summer of 2021, 15 out of 16 events that occurred in Sifnos also occurred in Lesvos (280 km upwind). Considering that these events typically lasted for more than 4 hours they appear to take place on a regional scale. However, additional measurements in the Aegean are needed to confirm that this is indeed happening. Kalkavouras et al. (2017) analyzing NPF events in Santorini, a nearby island,
during a short-term summer campaign (15 to 28 July 2013), found that under the Etesian flow their spatial extent can be at least 250 km. However, the fact that the NPF frequency in Sifnos is almost half than that in Lesvos, suggests that as the air masses travel over the sea, the concentrations of the gaseous precursors are getting progressively lower, resulting in fewer events.

All the three weak NPF events in Chania occurred when the air was coming from the north (Fig. S10). In all cases
the trajectories passed over land areas of Eastern Evia and Attika regions about 10 hours before arriving at the site. Chania were also exposed to the northeastern flow that appears to favor NPF in Sifnos and Lesvos, but a very low NPF frequency and weak events were observed under these conditions.

The majority (69%) of the events in Finokalia were also connected with northern and northeastern air masses (Fig. S11). Out of the 16 events occurred under northeastern flow, 11 also took place in Sifnos and Lesvos, suggesting regional
NPF. An important fraction (31%) of the events in Finokalia, occurred when the wind was coming from the west. The air masses in these cases had spent 2-6 hours over mainland Crete before arriving at Finokalia, unlike Chania (145 km west) where they arrived directly at site after spending >12 h over sea. These observations (including the five remaining events with northeastern air masses that occurred only in Finokalia) suggest that NPF events in relatively small spatial scale (less than 150 km) are possible in this area. These "local" events were estimated to account for 36% of the total. An example of such an event
is shown in Fig. S12. Our explanation about these "local" events in Finokalia, is that the air masses that come both from the northeast shift and pass over Crete (Kalkavouras et al., 2021; Rizos et al., 2022) picking up ammonia that is emitted in the agricultural area of central Crete.

### 4.5.3 Western and Central Greece

The sites in the western and central parts of the country (PAT[SUB], NEO[RUR], IOA[SUB], and ATH[SUB]) had all a NPF frequency of
less than 20%. During the measurement period, clean air masses from the western sector (Fig. 8c), typically spending more

than 12 h over the Ionian Sea, were arriving at the western sites (IOA[SUB], PAT[SUB] and NEO[RUR]). In Athens air masses from the northeast prevailed (during 45% of the days) while northern, western, and southern directions were also observed.

In Patras during the two strongest Class I events air masses, passed over the agricultural region of Ilia, south of Patras. 31% of the days were classified as transported events (Aktypis et al., 2023) with the estimated region of NPF being an agricultural area 150 km northeast of Patras.

Two thirds of the NPF events (including all 3 Class I events) occurred in Athens when the air masses were coming from the west or the northeast (Fig. S14). In all cases, they passed over the greater Viotia region (which has higher $NH_3$ emissions due to agricultural activity). It is worth mentioning that, excluding the NPF events, the western sector was not a frequently traversed pathway for the air masses (Fig. 8c). The other 33% were weak class II events which occurred when the wind was coming from the N-NE and passed over central Evia, where NPF could also be happening.

In Ioannina, NPF was favored by air masses passing from N. Macedonia and Albania (Fig. S15). The only two Class I events observed in Ioannina occurred in air masses originating from N. Macedonia and also passing over the Ptolemaida region and the agricultural area of Thessaloniki. Air masses from land areas of Albania were also associated with (weaker) events. Although Ioannina is in close proximity to the various sources of $SO_2$ and $NH_3$, the mountain range of Pindos, that separates western from eastern Greece, limits the passage of the air masses from the northeast. The lack of gaseous precursors together with the high condensation sink observed there and the meteorological conditions (high RH, air stagnation), could be part of the explanation why NPF is infrequent in this area.

Two out of the three NPF events observed in NEO occurred in northeastern air masses, passing over the Peloponnese before arriving at the site. In NEO, there were at least three cases where nucleation mode particles appeared during afternoon and nighttime hours and also showed signs of growth to larger sizes (Fig. S2). The trajectory analysis for these days revealed that these particles could be a result of transport from the Ilia agricultural region, located 100 km north. This region has also been found to affect NPF in Patras, when the air passes over it (Aktypis et al., 2023).

### 4.5.4 High elevation site

The air masses during 70% of the NPF events in Helmos were northeastern, while the corresponding fraction for the whole measurement period was 49%. The rest 30% of the events occurred when the air came either from the southwest or the northwest. The NE air masses passed over the greater agricultural area of Viotia a few hours before arrival. This is the same area that appears to affect the nearby stations, Patras, and Athens. There were also many cases (about 45% of all days) where distinct particle modes in the nucleation and Aitken range appeared during early afternoon (around 14:00 LT). These modes did not grow and were probably the result of transported events (traditionally classified as undefined). This lack of further growth is probably due to the limited levels of condensable vapors in this high elevation area.

## 4.7 Concurrent events in multiple sites

There was not a single day when NPF events were observed at all stations. This is reasonable considering the topographic heterogeneity of Greece, and that each station was exposed to different air masses. NPF events took place at most (50-85%) of the sites during 10 days. During seven out of the ten cases, the predominant wind direction was from the northeast. Two cases occurred when the air masses originated from the west (Lesvos and Thrace continued to be affected by eastern air masses in one case and also had NPF). The last case occurred when the air masses originated from the north (passing over North Macedonia and Bulgaria) in all stations. Although the condensation sink in most cases was lower than the corresponding average of each station for the whole period, it was not statistically significant (a=0.01). It was significantly lower by 50% only during the two cases when the air masses were arriving from the west, spending many hours over the Ionian Sea. Being primarily of marine origin, these air masses were relatively clean, but appeared to have enough gaseous precursors in order for new particles to be formed. This can be explained by the fact that in all cases they spent some hours over land areas before arriving at each site and did not come directly from the sea.

The rarity of simultaneous events occurring in over half of the sites during the measurement period highlights the diverse nature of the atmosphere across the different locations in terms of its availability in NPF precursors.

## 5. Conclusions

Two summers (2020 and 2021) of extensive continuous particle number size distribution measurements in 11 different locations in Greece were analyzed to quantify the NPF frequency and its spatial extent across the country and also assess the factors affecting them.

A significant variability of the NPF frequency was observed among the different sites, varying from close to zero in the southwestern parts of the country to more than 60% in the northern central and eastern regions. An average particle growth rate of $5.7 \pm 2.9$ nm h$^{-1}$ was calculated for all sites and varied from 3.4 nm h$^{-1}$ in Mt. Helmos to 8 nm h$^{-1}$ in Finokalia. The average CS of events and non-events was similar in most of the measurement sites (6 out of 11).

The southern and western parts of the country, where NPF was infrequent, were characterized by lower and more localized ammonia emissions. Although the emissions of biogenic volatile organic compounds were higher in these areas, they did not appear to lead to enhanced new particle formation. The detailed analysis of the air mass trajectories revealed that the infrequent clear NPF events in these sites occurred when the air masses had spent a few hours over land areas with agricultural activities, and thus elevated ammonia emissions. Air masses arriving directly from the sea were not connected with NPF in all sites. The high NPF frequency sites in the north were in close proximity to large agricultural areas with the highest ammonia emissions of the country (higher than 2 kg m$^{-2}$ d$^{-1}$). The corresponding air masses during the NPF events in these areas, were also enriched with emissions from various big coal-fired power plants located in northern Greece and the neighboring countries. Many regional events were observed in the sites of Lesvos, Sifnos and Finokalia, all located in the Aegean Sea.

However, in Finokalia there were also events (36%) that were not observed in the nearby stations, making them more localized (<150 km). This supports the hypothesis that a compound is missing (probably ammonia) from those air masses, which after their passing over central Crete, they collect it increasing this way their potential to form new particles on a smaller spatial scale.

There was not a single day where NPF events took place in all the measurement sites. Simultaneous events occurred in at least half of the sites in only 10 out of 124 days. In most of the cases, northeastern air masses were arriving at the sites. The fact that concurrent events were extremely rare throughout the measurement period, signifies the heterogeneity of the atmosphere in the different locations regarding the availability of NPF precursors.

**Acknowledgements.** This work has received funding support by Greece and the European Union (European Regional Development Fund) via the project PANhellenic infrastructure for Atmospheric Composition and climatE chAnge (PANACEA, MIS 5021516). Part of this work was also supported by the RI URBANS project (grant agreement No 101036245 from the European Union's Horizon 2020 research and innovation program). Finally, we thank Prof. K. Kourtidis and Prof. N. Hatzianastassiou and G. Kostenidis for helping with the measurements, I. Kioutsioukis for providing the WRF meteorological data, Prof. R. Krejci for providing the data in NEO 2020, and Prof. A. Nenes for providing the PSM for the measurements in Patras.

**Author contributions.** AA was responsible for the summer 2021 measurements in Patras, Xanthi and Ioannina, analyzed the data and wrote the paper, DP performed the stimulations and calculated the emissions of $SO_2$, $NH_3$, and organics, CK, AM, CV and KF helped with the measurements in Patras, EK helped with the measurements in Thrace, PK, AB and NM were responsible for the measurements in Lesvos, Sifnos, NK was responsible for the measurements in Finokalia, KE, SV and MIG were responsible for the measurements in Athens and Mt. Helmos, AK and CS were responsible for the measurements in Thessaloniki, ML and SEC were responsible for the measurements in Chania and SNP was responsible for the design of the study, the synthesis of the results and contributed to the writing of the paper.

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

**Table 1: Instrumentation, size range and temporal resolution for each measurement site during the 2020 and 2021 summer campaigns.**

| Observable | Instrument/ Model | Size range (nm) | Resolution | Period |
|---|---|---|---|---|
| **Athens (ATH – Suburban)** | | | | |
| Size distribution | SMPS (Classifier 3080, DMA 3081, CPC 3022A, TSI) | 10 – 550 | 5 min | 2020 and 2021 |
| Black carbon | Aethalometer AE33, Magee Scientific | | 1 min | 2020 and 2021 |
| **Thessaloniki (THES - Urban)** | | | | |
| Size distribution | SMPS (model 3034, TSI) | 10 – 487 | 5 min | 2020 and 2021 |
| **Patras (PAT - Suburban)** | | | | |
| Size distribution | SMPS (Classifier 3080, DMA 3081, CPC 3787, TSI) | 14 - 750 | 4 min | 2020 and 2021 |
| | SMPS (Classifier 3080, DMA 3085, CPC 3775, TSI) | 4 - 65 | 4 min | 2021 |
| | PSM (A11 nCNC, Airmodus) | 1.3 - 4 | 4 min | 2021 |
| $NO_x$, $NH_3$ | $NO_x$ and $NH_3$ analyzer T201, Teledyne | | 1 min | 2020 and 2021 |
| $SO_2$ | $SO_2$ analyzer 43i-TLE, Thermo Scientific | | 1 min | 2020 and 2021 |
| CO | CO analyzer API 300E, Teledyne | | 1 min | 2020 and 2021 |
| Black carbon | Aethalometer AE33, Magee Scientific | | 1 min | 2020 and 2021 |
| **Ioannina (IOA - Suburban)** | | | | |
| Size distribution | SMPS (Classifier 3080, DMA 3081, CPC 3775, TSI) | 14 – 723 | 3 min | 2020 and 2021 |
| **Thrace (THR - Rural)** | | | | |
| Size distribution | SMPS (Classifier 3080, DMA 3081, CPC 3787, TSI) | 14 - 723 | 3 min | 2020 and 2021 |
| **Lesvos (LES - Remote)** | | | | |
| Size distribution | SMPS (custom built, CPC 3786, TSI) | 9 – 430 | 5 min | 2021 |
| **Sifnos (SIF - Remote)** | | | | |
| Size distribution | SMPS (model 3034, TSI) | 10 – 487 | 5 min | 2020 and 2021 |
| Black carbon | Aethalometer AE33, Magee Scientific | | 1 min | 2020 and 2021 |
| **Finokalia (FIN - Remote)** | | | | |
| Size distribution | SMPS (custom built TROPOS type, CPC 3772, TSI) | 9 – 850 | 5 min | 2020 and 2021 |
| Black Carbon | Aethalometer AE33, Magee Scientific | | 1 min | 2020 and 2021 |
| **Chania (CHA - Urban)** | | | | |
| Size distribution | SMPS (Classifier 3082, DMA 3081, CPC 3775, TSI) | 12 – 593 | 5 min | 2020 and 2021 |
| **Methoni (NEO - Rural)** | | | | |
| Size distribution | DMPS (custom built, CPC 3772, TSI) | 10 – 806 | 15 min | 2020 |
| **Mt. Helmos (HAC - Remote)** | | | | |
| Size distribution | SMPS (custom built TROPOS type, CPC 3772, TSI) | 10 – 800 | 5 min | 2020 and 2021 |

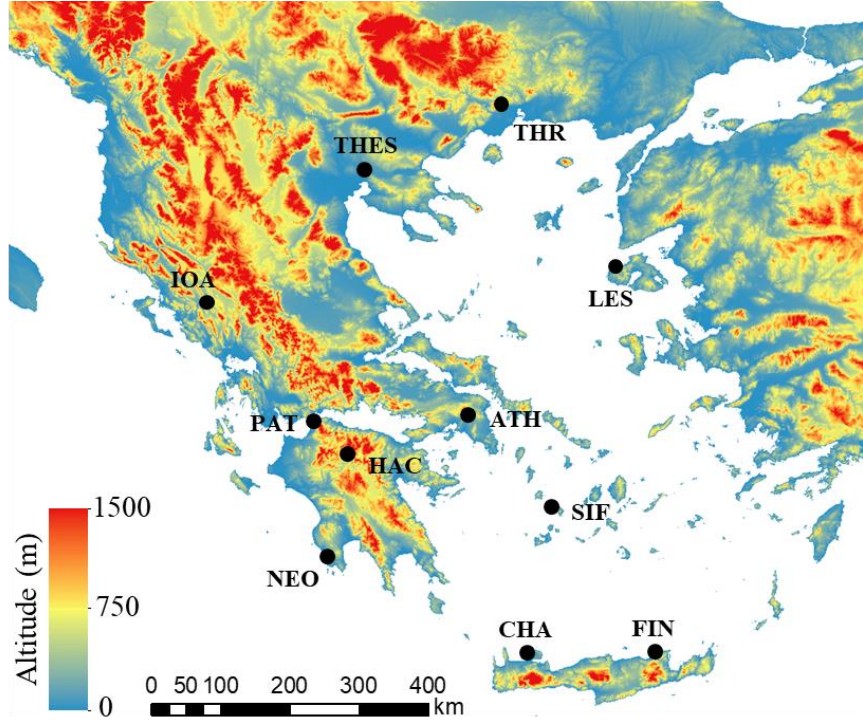

**Figure 1: Elevation map of Greece indicating the 11 measurement sites (map created using the European Digital Elevation Model (EU-DEM) data, by Copernicus-EEA for Eastern Europe).**

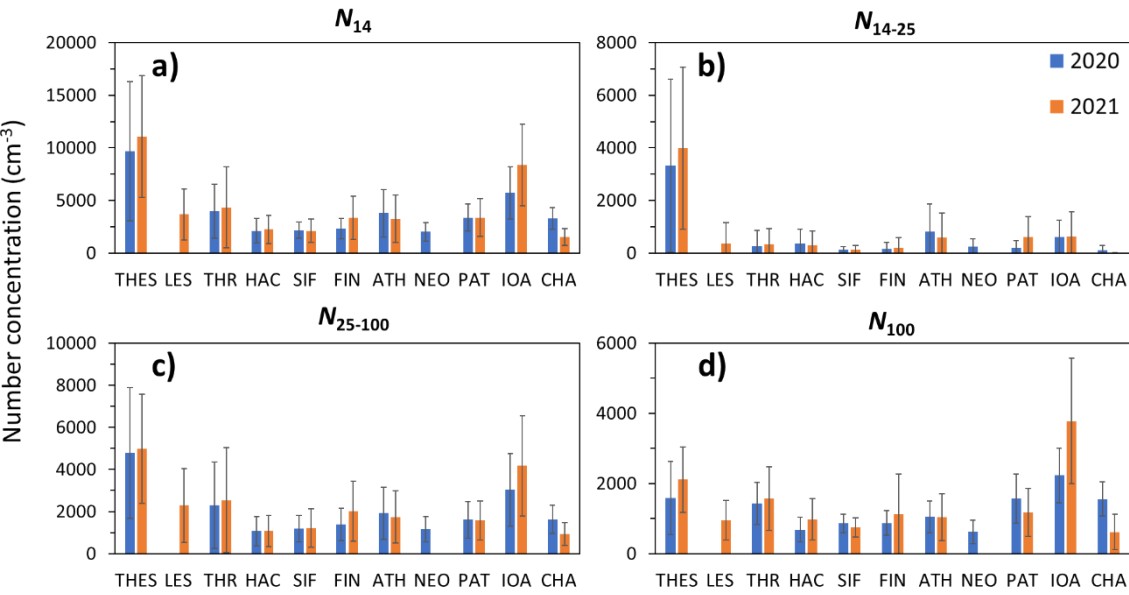

**Figure 2: The average particle number concentrations of a) the $N_{14}$, b) the nucleation mode (14-25 nm), c) the Aitken mode (25-100 nm) and d) the accumulation mode (>100 nm) in each measurement site during the summer campaigns of 2020 and 2021. The error bars represent one standard deviation.**

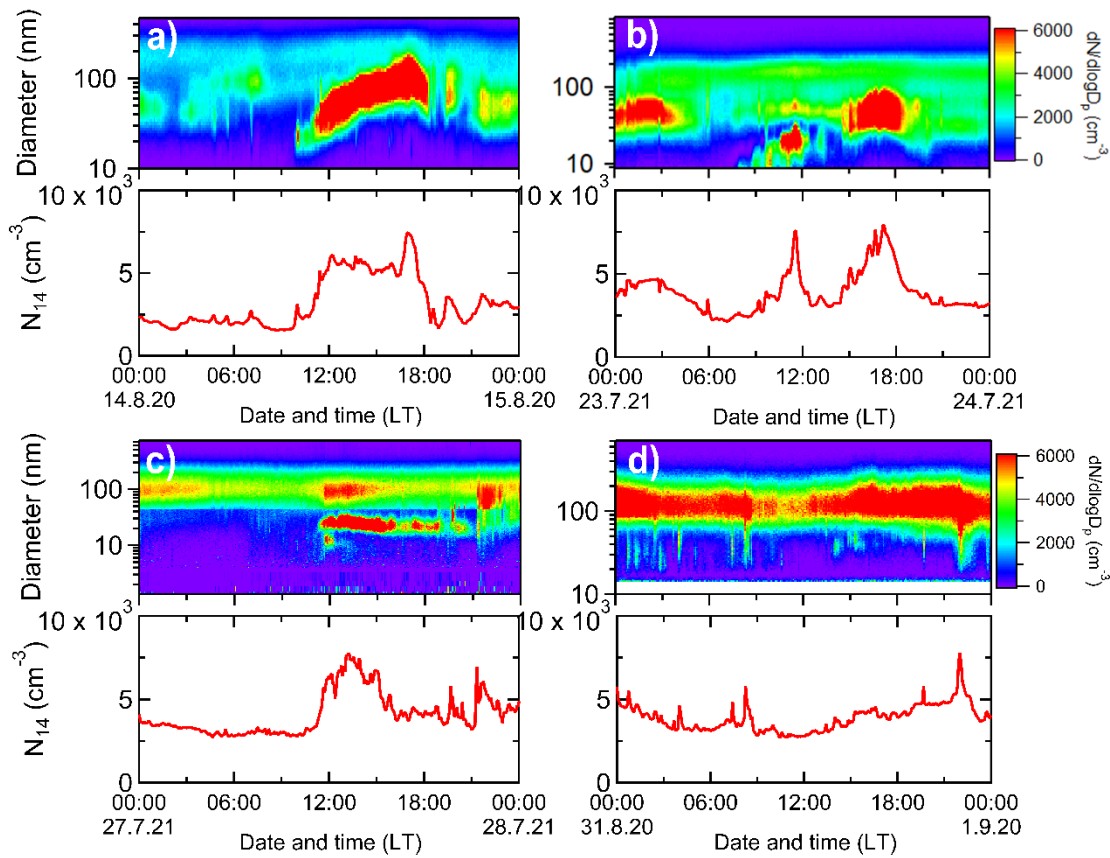

**Figure 3: Typical examples of a) a Class I NPF event occurred in Sifnos at 14/8/2020, b) a Class II NPF event occurred in Finokalia at 23/7/2021, c) an undefined event occurred in Patras at 27/7/2021 and d) a non-event in Chania at 31/8/2020. The corresponding $N_{14}$ during each day is also shown.**

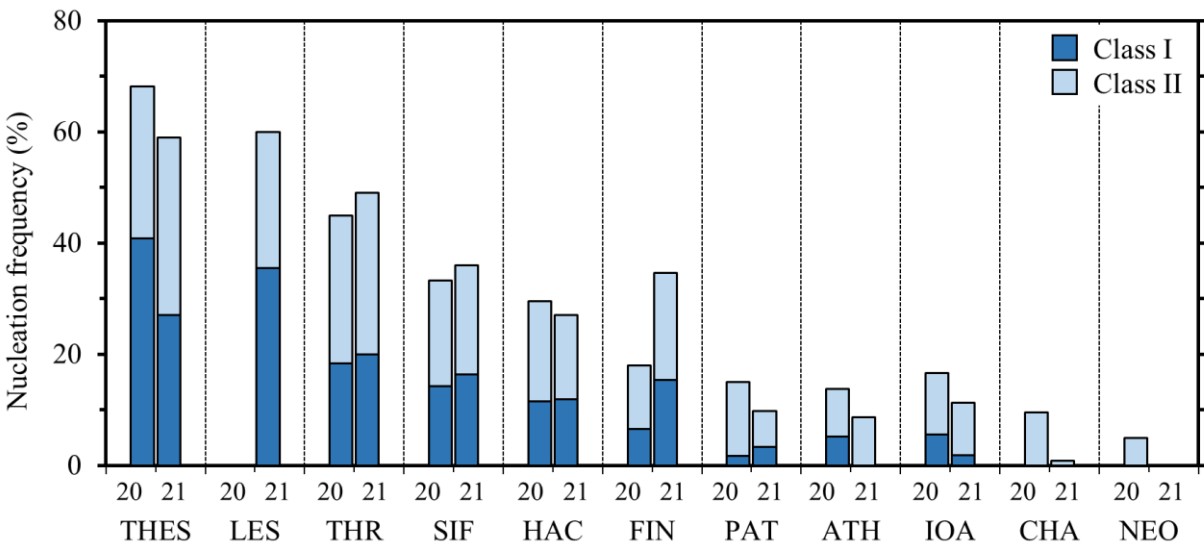

**Figure 4: The frequency of Class I and Class II events observed in each measurement site for the summer campaigns of 2020 and 2021.**

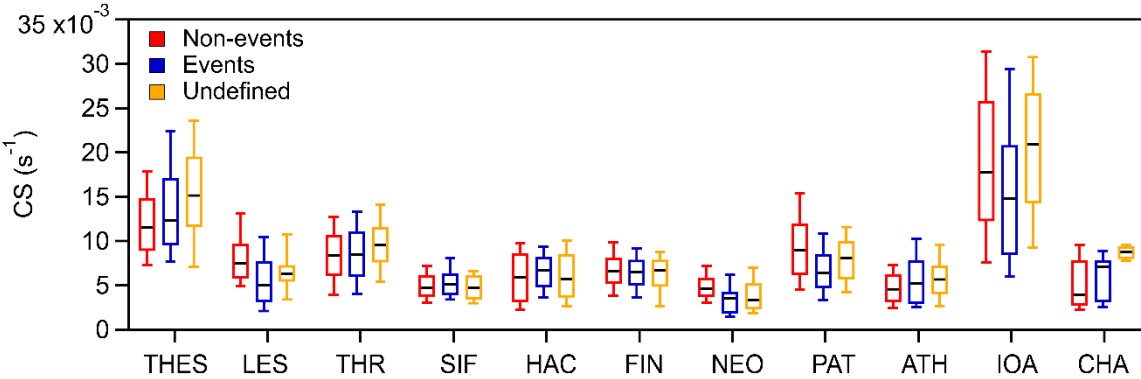

**Figure 5: Comparison of the average condensation sink (CS) values in the time period between 8:00 and 18:00 LT during NPF event, non-event, and undefined days in each station. The black lines represent the median, while the box edges the 25th and 75th percentiles. The whiskers correspond to the 10th and 90th percentiles.**

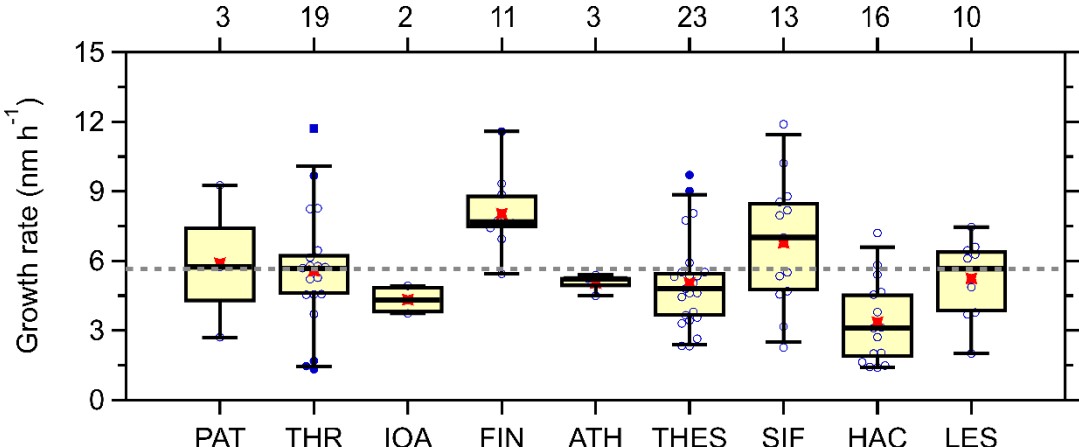

**Figure 6: Average growth rates of the NPF events in each station (where Class I events were observed). The black lines represent the median, the red points the average, while the box edges the 25th and 75th percentiles. The whiskers correspond to the 10th and 90th percentiles. The data points are shown with blue circles. The unfilled circles are data points that lie in between the 10-90% range while the filled ones are outliers. On the top, the number of data points is shown.**

865

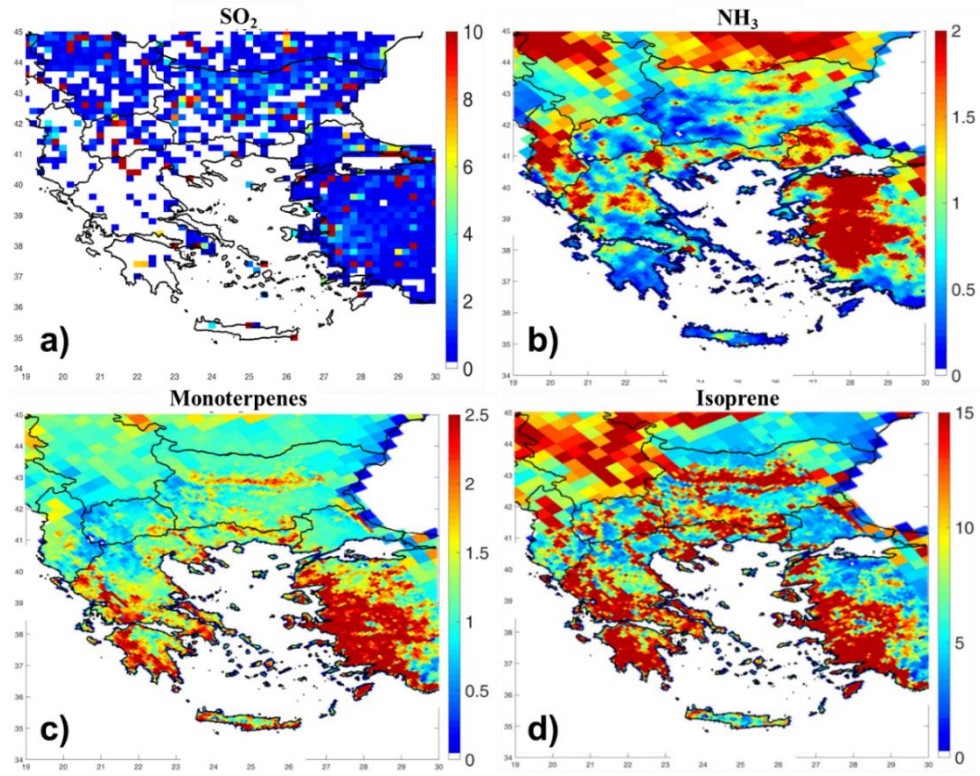

**Figure 7: Emissions (in kg km⁻² d⁻¹) of a) SO₂, b) NH₃, c) monoterpenes and d) isoprene in Greece during the summer of 2020.**

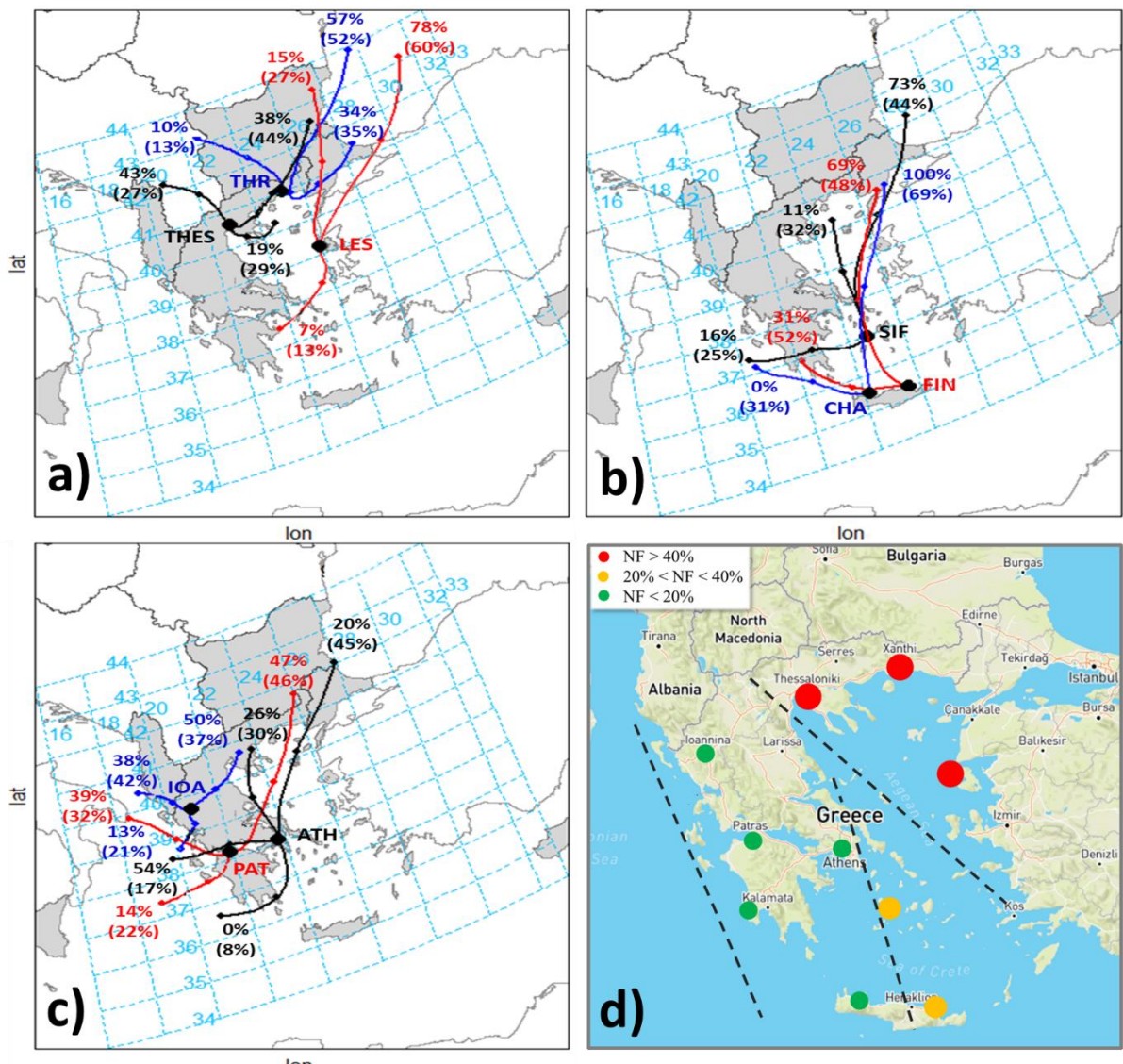

**Figure 8: Cluster frequencies during the NPF events at: a) northeastern Greece, b) South Aegean and c) the western and central Greece. The numbers in the parentheses show the corresponding frequency of the trajectories over the whole studied period (created by R version 4.1.1 using the Openair (Carslaw and Ropkins, 2012) version 2.16-0 and the MapData 2022 packages). The spatial distribution of the NPF frequency across the country (d) is also depicted (map from © Google Maps using © OpenStreetMap contributors 2022 (Distributed under the Open Data Commons Open Database License (ODbL) v1.0) and © Mapbox and modified by the authors).**