# Peer review of "Significant spatial gradients in new particle formation frequency in Greece during summer"

_EGUsphere, 2023_

## Referee Comment (RC1)

Aktypis et al analyzed particle number size distribution measurements from two summer campaigns in 11 sites in Greece. Their findings indicate that new particle formation (NPF) frequency in Greece is very heterogeneously divided, with areas in the northeast of Greece showing both higher average emissions and higher frequency of NPF than in the souther and western part of Greece. Their findings connect the higher NPF frequency with higher anthropogenic emissions in the prevailing air masses in the area and suggest that ammonia and/or amines could be the limiting factor for NPF in the area. This shines light into the possible NPF mechanisms in the Eastern mediterranean and characterizing the spatial extent of NPF is an important new contribution into literature. The paper is well written, and the methodology is sound, but the paper would benefit from some clarifications regarding both terminology and conclusions. Therefore, I believe it can be recommended for publication in ACP after minor revisions.

**General Comments**

Line 40: "Atmospheric nucleation and… " (also other places, e.g. line 32, Fig 4 caption). Recent studies suggest that new particle formation may not technically always require nucleation (overcoming an energy barrier) as the initial cluster are already stable. Also, since your measurements start mainly at 14 nm, you cannot really conclude about the initial steps of particle formation. Therefore, I would generally be careful of using the term "nucleation" here and other places in the manuscript.

Line 103: "The sites were classified as urban, suburban, and rural according to the criteria proposed by Larssen et al. (1999)" This classification is not easily identifiable from the text. The classification could be added to Table 1.

Line 140: The assumption that Xanthi and Xylagani are both representative of the broader Thrace region is poorly explained. Wouldn't a location with a major road nearby and a peri-urban location have more anthropogenic influence in comparison to a smaller rural site?

Line 200: What is your criteria for a weak NPF event?

Line 241: You cannot observe the full nucleation mode if your measurements starts from 14 nm. Do you mean that when the observed geometric mean diameter started increasing? Note, that this might significantly differ from the actual starting time of the new particle formation process, as the growth rates are quite low.

Line 259 (and elsewhere e.g. 313). By word average, do you refer to mean or median? Over what time period was the average calculated?

Lines 266-269: I'm not sure if it is meaningful to calculate and compare the fractions of nucleation mode particles, since you are measuring only the upper part of the nucleation mode, and also because the upper limit of the measurements varies between sites. Have you estimated how does the different upper size limits for the DMPS/SMPS systems affect the N14 concentration?

Line 330: Do PM2.5 and CS correlate? Your calculated CS does not extend up to 2.5 micrometers.

Line 333: The average wind speed was low also in THR, with similar wind direction, yet the average CS is much lower. Why is the air stagnation only relevant to Ioannina?

Chapter 4.3. It is interesting that the GR varied so little, even though the NPF frequency varied considerably. Do the GRs show any dependency on meteorological conditions (e.g. temperature) at those sites where you have larger amount of data?

Line 367: "Emissions of amines in these areas may also be elevated" Could you elaborate on this?

Line 377: Do you think the warm temperatures during your measurement period could also affect the volatility distribution of the oxidized organics, preventing the formation of least volatile compounds that can form particles and participate in the early steps of growth (see e.g. doi.org/10.5194/acp-20-9183-2020)?

Line 416: Sifnos and Lesvos are mostly surrounded by sea. Considering that the sea likely contributes very little to the growth of particles and considering the time it takes for the airmass to travel between the two sites, it is not directly evident that you can call these events regional, even though they happen at the same time. I believe that whether these can be called regional events should be discussed more in the article.

**Technical Comments**

Line 52: ….the dominating  new particle formation mechanisms….

Line 200: missing y (study)

Line 266: I do not understand how you have arrived at these numbers and variances. 12%+9% does not come to 35%, nor does 54%+5% come to 62%.

Line 386: Cluster analysis is a very general term and openAir is a big R package. Can you specify what kind of cluster analysis you did?

Line 447: "Unlike the NPF events…" This is an unclear sentence, please clarify. Do you mean that that outside NPF event days, the western pathway was not frequently traversed?

Figure 2: Describe what the whiskers mean.

Figure 3: Subplots a) and d) only have one value on the y-axis. It would be better to have at least two values visible for easier reading.

Figure 6: What is the difference between filled blue circles and unfilled blue circles? Please explain this in the caption.

Figure S2: The caption here appears to belong to Figure S1.

---

## Author Response (AR1)

**Responses to the Comments of the Reviewers**

**Reviewer #1**

(1) Aktypis et al analyzed particle number size distribution measurements from two summer campaigns in 11 sites in Greece. Their findings indicate that new particle formation (NPF) frequency in Greece is very heterogeneously divided, with areas in the northeast of Greece showing both higher average emissions and higher frequency of NPF than in the southern and western part of Greece. Their findings connect the higher NPF frequency with higher anthropogenic emissions in the prevailing air masses in the area and suggest that ammonia and/or amines could be the limiting factor for NPF in the area. This shines light into the possible NPF mechanisms in the Eastern Mediterranean and characterizing the spatial extent of NPF is an important new contribution into literature. The paper is well written, and the methodology is sound, but the paper would benefit from some clarifications regarding both terminology and conclusions. Therefore, I believe it can be recommended for publication in ACP after minor revisions.

We appreciate the positive assessment of our work by the reviewer. Our answers (in regular font) follow each comment of the reviewer (in blue).

**General Comments**

(2) Line 40: "Atmospheric nucleation and…" (also other places, e.g. line 32, Fig 4 caption). Recent studies suggest that new particle formation may not technically always require nucleation (overcoming an energy barrier) as the initial cluster are already stable. Also, since your measurements start mainly at 14 nm, you cannot really conclude about the initial steps of particle formation. Therefore, I would generally be careful of using the term "nucleation" here and other places in the manuscript.

We agree with the comment of the reviewer. The scope of this work was to provide an overall picture of the importance of the formation of new particles in the atmosphere of different areas in Greece, and how its frequency can be affected by anthropogenic and/or natural emissions. The term "nucleation" was replaced by "new particle formation or NPF" in the revised manuscript.

(3) Line 103: "The sites were classified as urban, suburban, and rural according to the criteria proposed by Larssen et al. (1999)" This classification is not easily identifiable from the text. The classification could be added to Table 1.

We have added this information to Table 1 (in parenthesis after the name of the site) following the suggestion of the reviewer.

**(4)** Line 140: The assumption that Xanthi and Xylagani are both representative of the broader Thrace region is poorly explained. Wouldn't a location with a major road nearby and a peri-urban location have more anthropogenic influence in comparison to a smaller rural site?

Although the two sites can have significant differences regarding local ultrafine particle emissions, the study focuses on NPF events (lasting for >2 h) that take place on a spatial scale larger than the distance (about 45 km) between the two sites. In addition, the fine PM and particle concentration levels of the two sites were compared and were found to be similar. Finally, the HYSPLIT analysis revealed that both areas are generally affected by the same air masses (mostly north and northeast) which is important when studying NPF. These points are now clarified in the revised manuscript.

**(5)** Line 200: What are your criteria for a weak NPF event?

The term "weak" was used here (and elsewhere, like in Chania) to describe events when the new particles grew in size, but either their number concentration was quite low, or their growth was limited. The term has also been used by Lee et al. (2020). This term is now clarified in the revised manuscript.

**(6)** Line 241: You cannot observe the full nucleation mode if your measurements start from 14 nm. Do you mean that when the observed geometric mean diameter started increasing? Note, that this might significantly differ from the actual starting time of the new particle formation process, as the growth rates are quite low.

We agree that this statement was obscure. We now clarify that the starting time of an event was identified when the observed (with the data available) nucleation mode geometric diameter started increasing. This was necessary for the calculation of the growth rates. For this reason, we did not attempt to determine the actual starting time of the new particle formation process.

**(7)** Line 259 (and elsewhere e.g., 313). By word average, do you refer to mean or median? Over what time period was the average calculated?

In line 259, the word "average" refers to the mean number concentration (in the corresponding size ranges), calculated for both campaigns in each station. It aims to provide a general idea about the particle number concentrations in Greece during the measurement periods. In line 313 (and elsewhere, where we compare parameters like CS, RH, wind speed) we again use the mean values, but calculated in the time window between 8:00-18:00 LT, when NPF generally occurs. This information is now provided in the revised paper.

**(8)** Lines 266-269: I am not sure if it is meaningful to calculate and compare the fractions of nucleation mode particles, since you are measuring only the upper part of the nucleation

mode, and also because the upper limit of the measurements varies between sites. Have you estimated how does the different upper size limits for the DMPS/SMPS systems affect the $N_{14}$ concentration?

We do clarify in the revised paper that the reported number concentration for the nucleation mode does not include all the corresponding particles. We do specify the lower size of the distributions used to avoid misunderstandings. The upper limit varies from 430 to 800 nm in the various sites. The particle number concentration above 430 nm is negligible compared to the particle number in the range of 14-430 nm in all sites. More specifically, we calculated the ratio $N_{430\text{-}800}/N_{14}$ (contribution of the particles in the range 430 – 800 nm to the $N_{14}$) for all sites and it was always lower than 2%. This information has been added to the text.

**(9)** Line 330: Do PM$_{2.5}$ and CS correlate? Your calculated CS does not extend up to 2.5 micrometers.

For typical continental and urban conditions most of the CS is below 1 μm (Seinfeld and Pandis, 2016). As a result, it tends to correlate will with PM$_{2.5}$ because they are both determined mainly by the accumulation mode. Exceptions, for Greece, are periods with Sahara dust episodes. The surface area between 1 and 2.5 μm for these periods without very high dust levels is usually a small fraction of the total.

**(10)** Line 333: The average wind speed was low also in THR, with similar wind direction, yet the average CS is much lower. Why is the air stagnation only relevant to Ioannina?

We agree with the reviewer that air stagnation alone does not explain the high CS observed in Ioannina. It is the combination of the low wind speed with significant local sources that leads to the high average CS. This is not the case in Thrace. This point is now addressed on the revised manuscript.

**(11)** Chapter 4.3. It is interesting that the GR varied so little, even though the NPF frequency varied considerably. Do the GRs show any dependency on meteorological conditions (e.g., temperature) at those sites where you have a larger amount of data?

We investigated the potential linkages between meteorological conditions (T, RH, wind speeds) for the sites that had enough growth rate data points (THES, THR, FIN, SIF, HAC, LES). We did not observe any systematic dependency of the GR on any of these parameters. This suggests that other non-meteorological factors control the growth rate in these regions. This point is now discussed in the paper.

**(12)** Line 367: "Emissions of amines in these areas may also be elevated" Could you elaborate on this?

The most important sources of amines (animal husbandry, agricultural activities, food industry, etc.) have also been connected to elevated emissions of ammonia (Ge et al., 2011).

Given that the forementioned regions are the largest agricultural areas in Greece with also increased livestock, higher levels of amines are also expected there. This point is now made clear in the revised manuscript.

**(13)** Line 377: Do you think the warm temperatures during your measurement period could also affect the volatility distribution of the oxidized organics, preventing the formation of least volatile compounds that can form particles and participate in the early steps of growth (see e.g. doi.org/10.5194/acp-20-9183-2020)?

This is a good point, and we now include a brief discussion about this point. Indeed, the temperatures during the measurement period were high (especially during daytime, when NPF typically occurs). This can lead not only to more volatile oxidized organics but may also reduce the stability of small clusters in the atmosphere (Bousiotis et al., 2021). The relatively high temperatures might also be part of the explanation of why the organics do not appear to enhance the NPF frequency in the corresponding high emission areas.

**(14)** Line 416: Sifnos and Lesvos are mostly surrounded by sea. Considering that the sea likely contributes very little to the growth of particles and considering the time it takes for the airmass to travel between the two sites, it is not directly evident that you can call these events regional, even though they happen at the same time. I believe that whether these can be called regional events should be discussed more in the article.

Although the distance between the two sites (about 300 km) is large enough to characterize these events as regional (Kulmala et al., 2004; Kerminen et al., 2018), we agree that we cannot prove at this stage that they also occur in the area between the two islands. We have now added a small discussion, addressing the point of the reviewer about the uncertainty of the scale of these events.

**Technical Comments**

**(15)** Line 52: ….the dominating nucleation new particle formation mechanisms….

We have corrected the sentence to "…the dominating new particle formation mechanisms…" following the suggestions of the reviewer.

**(16)** Line 200: missing y (study).

We have corrected the typo in this sentence.

**(17)** Line 266: I do not understand how you have arrived at these numbers and variances. 12%+9% does not come to 35%, nor does 54%+5% come to 62%.

The numbers next to the average number concentration percentage of each mode, represent one standard deviation from the mean. This is now explained in the revised paper. These two cities were more than two standard deviations from the mean.

**(18)** Line 386: Cluster analysis is a very general term and openAir is a big R package. Can you specify what kind of cluster analysis you did?

In order to group all the trajectory endpoints and calculate the clusters we used an angle distance matrix. Compared to the Euclidean, the angle method takes more into account the movement patterns and shape of the trajectories. We now include a brief description about the method we used in the openAir package to calculate the clusters of the trajectories.

**(19)** Line 447: "Unlike the NPF events…" This is an unclear sentence, please clarify. Do you mean that outside NPF event days, the western pathway was not frequently traversed?

We have rewritten this rather confusing sentence.

**(20)** Figure 2: Describe what the whiskers mean.

A description of the whiskers was added to the caption of the figure following the reviewer's suggestion.

**(21)** Figure 3: Subplots a) and d) only have one value on the y-axis. It would be better to have at least two values visible for easier reading.

Figure 3 was improved by adding more values on the y-axis.

**(22)** Figure 6: What is the difference between filled blue circles and unfilled blue circles? Please explain this in the caption.

The filled blue circles are outliers (higher or lower than the 90% or 10% of all data points respectively). The unfilled blue circles are data points that lie in between the 10-90% range. This is now described in the figure caption of the revised manuscript.

**(23)** Figure S2: The caption here appears to belong to Figure S1.

The caption of the figure has been corrected.

**References**

Bousiotis, D., Pope, F.D., Beddows, D.C.S., Dall'Osto, M., Massling, A., Nøjgaard, J.K., Nordstrøm, C., Niemi, J.V., Portin, H., Petäjä, T., Perez, N., Alastuey, A., Querol, X., Kouvarakis, G., Mihalopoulos, N., Vratolis, S., Eleftheriadis, K., Wiedensohler, A., Weinhold, K., Merkel, M., Tuch, T., and Harrison, R.M.: A phenomenology of new particle formation (NPF) at 13 European sites, Atmos. Chem. Phys., 21, 11905–11925, 2021.

Ge, X., Wexler, A., S., Clegg, S., L.: Atmospheric amines – Part I. A review, Atmos. Envi., 45, 524-546, 2011.

Kerminen, V.-M., Chen, X., Vakkari, V., Petäjä, T., Kulmala, M., and Bianchi, F.: Atmospheric new particle formation and growth: review of field observations, Environ. Res. Lett., 13, 103003, 2018.

Kulmala, M., Vehkamäki, H., Petäjä, T., Dal Maso, M., Lauri, A., Kerminen, V.M., Birmili, W., and McMurry, P.H.: Formation and growth rates of ultrafine atmospheric particles: A review of observations, J. Aerosol Sci., 35, 143–176, 2004.

Lee, H., Lee, K., Lunder, C. R., Krejci, R., Aas, W., Park, J., Park, K.-T., Lee, B. Y., Yoon, Y. J., and Park, K.: Atmospheric new particle formation characteristics in the Arctic as measured at Mount Zeppelin, Svalbard, from 2016 to 2018, Atmos. Chem. Phys., 20, 13425–13441, 2020.

**Reviewer #2**

**(1)** The work by Aktypis et al. presents a very detailed spatial analysis of NPF in the area of Greece. The study, which includes the surprising number of 11 stations, focuses on the gradient of NPF frequency observed from west to east during summertime. The authors suggest that NPF events are limited by the presence of ammonia (and or amines), showcasing some of the sites to prove their point, on top of the overall gradient. The spatial scale of the events is also discussed, which is important because multi-site analysis concerning NPF are scarce in literature. The findings of this work are important, and the manuscript should be published after some minor revision is applied.
We appreciate the positive assessment of our work by the reviewer. Our answers (in regular font) follow each comment of the reviewer (in blue).

**Major comments**

**(2)** Even though the authors have put a big effort in placing evidence about the relation on ammonia with NPF, the effect of temperature should not be discarded. Have the authors checked the relation of NPF with temperature? I encourage the authors to discuss this topic.
We compared the average ambient temperature distributions (from 08:00 to 18:00) during event and nonevent days in all sites. For most sites (THR, SIF, FIN, NOE, PAT, IOA) the temperature during event days was statistically the same in during nonevent days. In THES, LES and CHA the temperature was on average a little higher (less than one degree) during NPF events, while the opposite was observed for HAC and ATH. This analysis suggests that temperature did not play a dominant role in determining the occurrence of NPF. This point is discussed in the revised paper and the corresponding figure showing the analysis results has been added to the supplementary information.

**(3)** Have the authors investigated NPF using the entire size range available in each site, or they focused only on the sizes where all sites were available, i.e. above 14 nm. This can make a difference especially if a site (e.g., Patras) has a much wider range than the others. Please clarify this point in the methods and discuss how the NPF may change if the size range was uniform at all sites.
Different size ranges can indeed introduce uncertainty in the comparison between the various sites, especially when studying NPF. The whole available size range at each site was utilized for the classification of the days. Excluding Patras, the lowest detectable size in all stations varied between 9 and 14 nm, which (although it introduces some uncertainty) makes the size distributions comparable for the classification step. In Patras, the classification was performed twice: a) by accounting both the full-size distributions and b) accounting only the ones for the size range 14 – 700 nm. The low frequency of NPF events in Patras was the same with both methods. Information about sub-10 nm in Patras was

valuable for the identification and interpretation of the undefined events (Aktypis et al., 2023), observed frequently in that area. A clarification and a brief discussion about the size ranges used for the analyses has been added to the Methods section of the revised manuscript.

**(4)** The author uses the term "nucleation" in the manuscript, even though the article refers to sizes over 10 nm for most sites. The term should be used cautiously.

We agree with the reviewer and the term "nucleation" was replaced by "new particle formation or NPF" in the revised manuscript.

**(5)** Fig 7b shows elevated ammonia emissions over Ioannina, but the nucleation frequency is low. How does this match with the general conclusion of this work?

The formation of new particles in the atmosphere is a complex phenomenon, affected by several parameters, including the availability of precursors (ammonia, sulfuric acid, organics etc.), the condensation sink and the meteorological conditions. Estimated ammonia emissions are elevated in the Ioannina area, but the NPF frequency is low. This could be partially explained by the increased mass concentration of preexisting particles in that area, that results in higher condensation sink (Fig. 5). The ambient CS can be further enhanced by the relatively high RH in the area due to the lake next to the city. A brief discussion about this interesting point has been added to the paper.

**(6)** Please take some time to explain to the readers your strategy for selecting to study summer periods only. It is apparent to me, but I suppose not to most of the readers of this work.

We focused on the summer periods because conditions favorable for NPF (high solar radiation, lower condensation sinks, higher biogenic volatile organic compound emissions (Nieminen et al., 2014; Bousiotis et al., 2021)) are present. This is a good period to investigate factors limiting new particle formation, because a high NPF frequency is expected everywhere. This is now explained in the paper.

**(7)** The readers put a lot of effort in placing their NPF frequency observations into the perspective of East Mediterranean. I suggest to add a comment on how this is related on a global scale as well, e.g. Nieminen et al., 2018 (https://doi.org/10.5194/acp-18-14737-2018)

A brief comparison of the results (frequency, growth rates during summer and the relation between those two) of this study and the observations of NPF on a global scale has been added to the revised manuscript.

**Minor comments**

**(8)** Statistical comparisons are done in the text but the method is not mentioned. T-test, ANOVA, Kruskal Wallis or another was used. Why compare at 99% CI? If the test was parametric did you take account the distribution shape? With what method?

A t-test was performed to compare the CS, RH and wind speed during the events and the non-events. The 99% confidence interval was chosen to reduce uncertainty in the comparisons. To evaluate the assumption that the data follow a normal distribution, the corresponding frequency and QQ plots were analyzed. Excluding outliers all distributions were close to normal. This information has been added to the paper.

**(9)** Lines 61-62: This comparison is obscure. Baalbaki et al., 2021 was investigating particles down to 1 nm. It is not clear which study you compare to? Pikridas et al., 2012 or Kalivitis et al., 2019? The latter concerns only particles down to 10 nm so a direct comparison is possible. See also comments above.

The comparison was made with the results of Pikridas et al. (2012). since their measurements (and thus their classification) included particles down to 1 nm. This is now clarified in the revised manuscript.

**(10)** Lines 80-82: This sentence has to be moved to the conclusions section or maybe rephrased to fit the introduction

The corresponding sentence has been rephrased to better fit in the Introduction section.

**(11)** Please add the information on the sampling site type in the header of every paragraph in Section 2.1., and optionally when addressing each site as a super/sub scipt. e.g. ATH$^{urb}$. For example in the Thrace region two very contradicting sites were used as representative of the larger area. The reader should be aware of the discrepancy.

We agree with the suggestion. We have now added the information in both the headers of each site's description and as a superscript next to each site's abbreviation (e.g., ATH$^{URB}$). Regarding Thrace, the difference between the two sites is now clarified in the text.

**(12)** Lines 149-150: The etesians affect the entire east Greece and not just the islands. Both Athens and Finokalia are affected for example. The statement in these lines is misleading in that sense.

We agree that this statement may be confusing for some readers, and we have rephrased it.

**(13)** Line 200 "Class II in this stud." Should be study.

We have corrected the typo.

**(14)** Lines 245-247: Also mention the temperature during expected NPF occurrence times, not just the campaign average.

The average temperatures during the expected NPF occurrence times for both years are now mentioned in the revised manuscript. These were calculated for the period between 9:00 and 15:00 LT when NPF typically occurs.

**(15)** Lines 283-285: Yes but in much higher latitudes. It is advised to mention Kalivitis et al., 2019 here as well.

The work of Kalivitis et al. (2019) is now mentioned in this part of the paper.

**(16)** The caption of Fig. 7 and of the description at line 349 do not match.

The emission inventory for the summer of 2018 (to our knowledge this is the latest validated inventory) was used to estimate the emissions for the summer of 2020. Here, there is the assumption that the emission sources have not changed so much between these years. Since $SO_2$, $NH_3$ and biogenic VOCs are typically emitted from sources that exhibit small changes with time, we expect the differences in the spatial distribution of these emissions between 2018 and 2020 to be minor. This is now clarified in the revised manuscript.

**(17)** Line 361-363: How is the conversion from $SO_2$ to $H_2SO_4$ done? Method and reference are missing.

The sulfuric acid concentration in the gas phase is assumed to be in pseudo-steady state and is proportional to the sulfur dioxide concentration, the OH radical concentration and inversely proportional to the condensational sink (Bardouki et al., 2003; Pikridas et al., 2012). This assumes that the reaction of sulfur dioxide with OH is the dominant source of sulfuric acid. The levels of OH were estimated using the predictions of the chemical transport model PMCAMx (Fountoukis et al., 2011) for the summer, while the sulfur dioxide and condensational sink were measured. This information has been added to the paper.

**References**

Baalbaki, R., Pikridas, M., Jokinen, T., Laurila, T., Dada, L., Bezantakos, S., Ahonen, L., Neitola, K., Maisser, A., Bimenyimana, E., Christodoulou, A., Unga, F., Savvides, C., Lehtipalo, K., Kangasluoma, J., Biskos, G., Petäjä, T., Kerminen, V.M., Sciare, J., Kulmala, M.: Towards understanding the characteristics of new particle formation in the Eastern Mediterranean, Atmos. Chem. Phys., 21, 9223–9251, 2021.

Bardouki, H., Berresheim, H., Vrekoussis, M., Sciare, J., Kouvarakis, G., Oikonomou, K., Schneider, J., and Mihalopoulos, N.: Gaseous (DMS, MSA, $SO_2$, $H_2SO_4$ and DMSO) and

particulate (sulfate and methanesulfonate) sulfur species over the northeastern coast of Crete, Atmos. Chem. Phys., 3, 1871–1886, 2003.

Bousiotis, D., Pope, F.D., Beddows, D.C.S., Dall'Osto, M., Massling, A., Nøjgaard, J.K., Nordstrøm, C., Niemi, J.V., Portin, H., Petäjä, T., Perez, N., Alastuey, A., Querol, X., Kouvarakis, G., Mihalopoulos, N., Vratolis, S., Eleftheriadis, K., Wiedensohler, A., Weinhold, K., Merkel, M., Tuch, T., and Harrison, R.M.: A phenomenology of new particle formation (NPF) at 13 European sites, Atmos. Chem. Phys., 21, 11905–11925, 2021.

Fountoukis, C., Racherla, P. N., Denier van der Gon, H. A. C., Polymeneas, P., Charalampidis, P. E., Pilinis, C., Wiedensohler, A., Dall'Osto, M., O'Dowd, C., and Pandis, S. N.: Evaluation of a three-dimensional chemical transport model (PMCAMx) in the European domain during the EUCAARI May 2008 campaign, Atmos. Chem. Phys., 11, 10331–10347, 2011.

Kalivitis, N., Kerminen, V.M., Kouvarakis, G., Stavroulas, I., Tzitzikalaki, E., Kalkavouras, P., Daskalakis, N., Myriokefalitakis, S., Bougiatioti, A., Manninen, H.E., Roldin, P., Petäjä, T., Boy, M., Kulmala, M., Kanakidou, M., and Mihalopoulos, N.: Formation and growth of atmospheric nanoparticles in the eastern Mediterranean: Results from long-term measurements and process simulations, Atmos. Chem. Phys., 19, 2671–2686, 2019.

Nieminen, T., Kerminen, V.-M., Petäjä, T., Aalto, P. P., Arshinov, M., Asmi, E., Baltensperger, U., Beddows, D. C. S., Beukes, J. P., Collins, D., Ding, A., Harrison, R. M., Henzing, B., Hooda, R., Hu, M., Hõrrak, U., Kivekäs, N., Komsaare, K., Krejci, R., Kristensson, A., Laakso, L., Laaksonen, A., Leaitch, W. R., Lihavainen, H., Mihalopoulos, N., Németh, Z., Nie, W., O'Dowd, C., Salma, I., Sellegri, K., Svenningsson, B., Swietlicki, E., Tunved, P., Ulevicius, V., Vakkari, V., Vana, M., Wiedensohler, A., Wu, Z., Virtanen, A., and Kulmala, M.: Global analysis of continental boundary layer new particle formation based on long-term measurements, Atmos. Chem. Phys., 18, 14737–14756, 2018.

Pikridas, M., Riipinen, I., Hildebrandt, L., Kostenidou, E., Manninen, H., Mihalopoulos, N., Kalivitis, N., Burkhart, J.F., Stohl, A., Kulmala, M., and Pandis, S.N.: New particle formation at a remote site in the eastern Mediterranean, J. Geophys. Res., 117, 1-14, 2012.